# The 8-17 DNAzyme can operate in a single active structure regardless of metal ion cofactor

Julia Wieruszewska [1,2], Aleksandra Pawłowicz[1,2], Ewa Połomska [1], Karol Pasternak[1], Zofia Gdaniec [1] & Witold Andrałojć [1] ✉

DNAzymes – synthetic enzymes made of DNA – have long attracted attention as RNA-targeting therapeutic agents. Yet, as of now, no DNAzyme-based drug has been approved, partially due to our lacking understanding of their molecular mode of action. In this work we report the solution structure of 8–17 DNAzyme bound to a $Zn^{2+}$ ion solved through NMR spectroscopy. Surprisingly, it turned out to be very similar to the previously solved $Pb^{2+}$-bound form (catalytic domain RMSD = 1.28 Å), despite a long-standing literature consensus that $Pb^{2+}$ recruits a different DNAzyme fold than other metal ion cofactors. Our follow-up NMR investigations in the presence of other ions − $Mg^{2+}$, $Na^+$, and $Pb^{2+}$ – suggest that at DNAzyme concentrations used in NMR all these ions induce a similar tertiary fold. Based on these findings, we propose a model for 8–17 DNAzyme interactions with metal ions postulating the existence of only a single catalytically-active structure, yet populated to a different extent depending on the metal ion cofactor. Our results provide structural information on the 8-17 DNAzyme in presence of non-$Pb^{2+}$ cofactors, including the biologically relevant $Mg^{2+}$ ion.

Nature has assigned the role of biological catalysts to proteins and to a lesser extent RNA molecules. In spite of this state of affairs conserved throughout biology, artificially engineered DNA catalysts (DNA enzymes; DNAzymes) have emerged in the 1990s[1,2]. The more inert chemical nature and inherent stability of such enzymes makes them most appealing for industrial and biomedical applications. DNAzymes whose catalytic activity is related to RNA processing (site-specific cleavage, ligation, phosphorylation, etc.) generate the most promise in the fields of biotechnology and medicine[3]. RNA-cleaving DNAzymes especially have already been widely applied as biosensors for targets, as diverse as, metal ions[4] up to whole bacterial cells[5]. However, ever since their discovery, the most anticipated use of RNA-cleaving DNAzymes is mRNA-level control in vivo. Unfortunately, DNAzyme application in therapy remains as of yet an unfulfilled promise, due to factors such as low concentration of their cofactors -- divalent metal ions − in cells or the difficulty to achieve multiple catalytic turnover in

these conditions[6,7]. Rationally redesigning DNAzyme molecules to amend for these shortcomings is currently a very difficult task, as our knowledge regarding their molecular mode of action remains limited. Only a deep structural and functional understanding of the most prominent RNA-cleaving DNAzymes can pave the path towards their successful therapeutic usage.

The 8-17 DNAzyme (Fig. 1A) is an archetypical RNA-cleaving DNA enzyme, first reported in 1997[2] and later repeatedly rediscovered in the presence of different metal ion cofactors[8]. Its constant reappearance in in vitro selections performed under very diverse conditions makes it the most versatile and pertinent DNAzyme identified to date. While originally discovered in the presence of $Mg^{2+}$[2], it was shown capable of using many other divalent ions as cofactors with the following trend of relative catalytic rates[8]: $Pb^{2+} \gg Zn^{2+} \gg Mn^{2+} \approx Co^{2+} > Cd^{2+} > Ni^{2+} > Mg^{2+} \approx Ca^{2+} > Sr^{2+} \approx Ba^{2+}$. 8–17 is also the most thoroughly investigated DNAzyme in terms of structure, sequential requirements, and reactivity[8].

[1]Institute of Bioorganic Chemistry, Polish Academy of Sciences, 61-704, Poznań, Noskowskiego 12/14, Poland. [2]These authors contributed equally: Julia Wieruszewska, Aleksandra Pawłowicz. ✉e-mail: wandralojc@ibch.poznan.pl

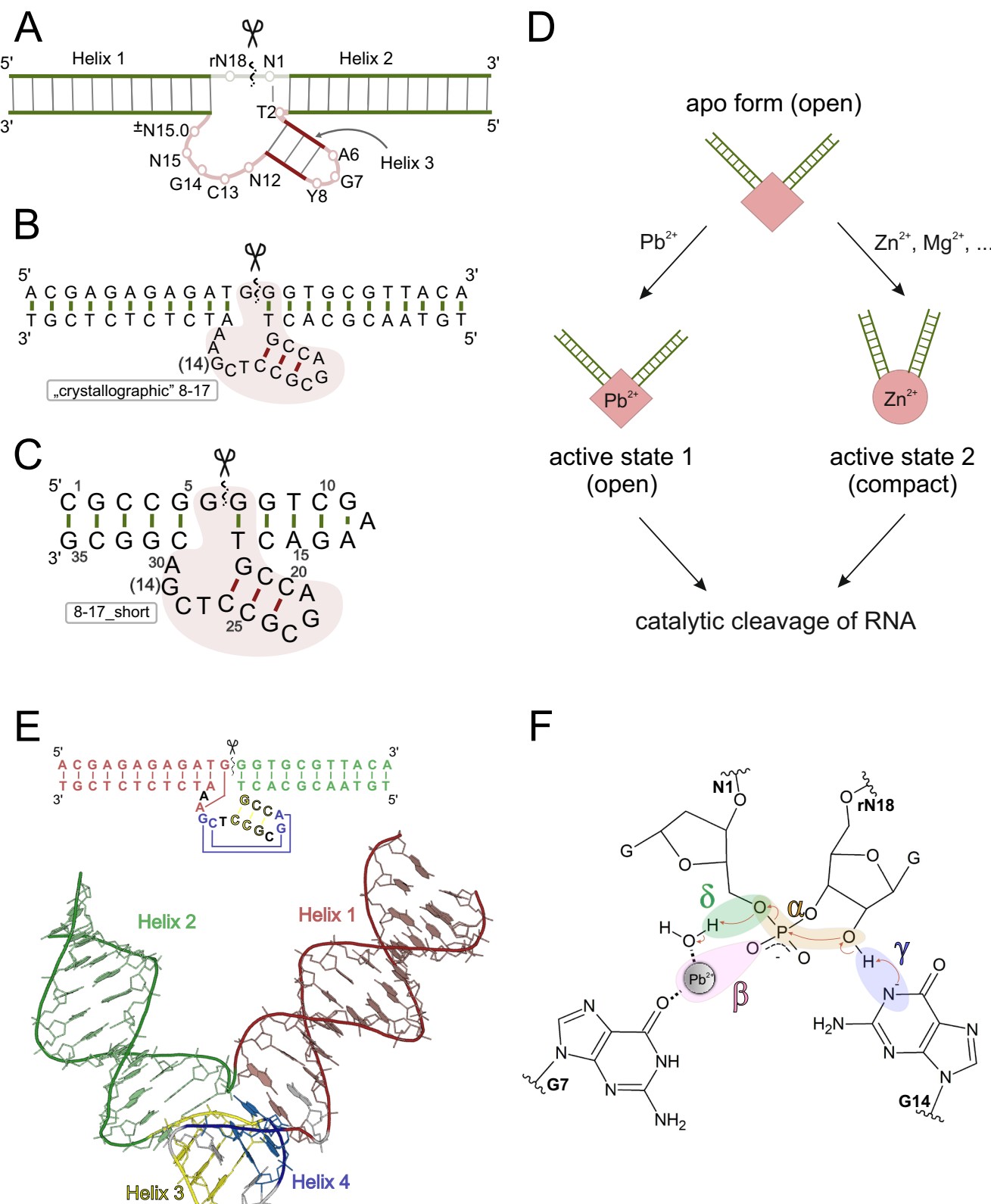

Fig. 1 | The 8-17 DNAzyme. A consensus structure, (B) construct used in the crystallographic study[13], (C) construct used in this NMR investigation, (D) current paradigm for 8–17 DNAzyme interplay with divalent metal ion cofactors, (E) the tertiary fold observed in Pb²⁺-bound crystallographic structure, (F) the proposed catalytic mechanism.

Structural studies using techniques such as FRET[9,10] and CD spectroscopy[11] uncovered a puzzling peculiarity in 8–17 DNAzyme's mode of action (Fig. 1D), by suggesting that it is capable to carry out its catalytic role in two distinct folded states, selected based on the cofactor present. Both techniques concluded that $Pb^{2+}$ binds directly to the apo form of the enzyme without substantially changing its open structure, and the catalytic reaction proceeds from this state. In contrast, in case of other cofactors - most prominently $Zn^{2+}$ and $Mg^{2+}$ - binding is accompanied by a large-scale conformational rearrangement within the DNAzyme towards a more compact structure and only this distinct fold supports catalysis with these cofactors. Thus, it appeared that in order to understand the 8–17 DNAzyme not one, but two catalytic mechanisms had to be elucidated, yet both of them long remained elusive in the absence of high resolution structural data[8].

While attempts at high-resolution structural characterization of DNAzyme molecules have long remained unsuccessful, recent years have brought major progress in the field. The first crystal structure of an RNA-ligating DNAzyme 9DB1 was reported in 2016[12]. This was followed by the crystal structures of the apo and $Pb^{2+}$ bound forms of 8–17 DNAzyme[13] (sharing the same overall fold; Fig. 1E) and most recently by the NMR structure of the 10–23 DNAzyme[14]. This last report has actually brought results that fulfill the promise of structure-informed DNAzyme engineering by proposing a single-atom substitution that rendered 10–23 several times more active[14]. Similarly, the crystal structures of 8–17 DNAzyme have reinvigorated the efforts to deeper understand its molecular mode of action. Based on the crystallographic report[13] and several follow-up experimental[15–18] and computational[19,20] studies a consensus regarding the catalytic mechanism in the presence of $Pb^{2+}$ started to emerge (Fig. 1F). First, the DNAzyme locks the nucleophile (2'OH group of r**N18**) and the scissile phosphate into a conformation that enables the in-line attack of the deprotonated nucleophile on the phosphorus atom[13]. Such a catalytic strategy is referred-to as α-catalysis[21,22]. In the crystal structure of 8–17 DNAzyme the entire tertiary fold appears to be stabilizing this particular conformation, with two substrate-binding arms containing the residues **N1** and **N18** forced into in a V-like orientation through co-axial stacking with two short helices present within the catalytic domain (Fig. 1E). The presence of one of those helices (Helix 4 in Fig. 1E) and the pseudo-knot it introduces within the catalytic domain, were only first made apparent in the crystallographic structure, leading to a previously unexpected intricate and rigid fold. Second, guanosine residue **G14** was identified[13,15] as the general-base that facilitates the deprotonation of the nucleophile (γ-catalysis). Third, the metal ion was in-turn proposed to assist in the protonation of the O5' leaving group (δ-catalysis) and/or to electrostatically stabilize the negatively charged transition state (β-catalysis). The $Pb^{2+}$ ion may also be implicated in the disruption of non-productive hydrogen bonds formed by the nucleophile in the pre-catalytic state[20]. Thus, 8-17 is now believed to utilize an intricate combination of different catalytic strategies to achieve its high RNA cleavage rates in the presence of $Pb^{2+}$, with its active site following the L-platform architecture found in multiple RNA-cleaving nucleic acid enzymes[23].

However, it remains unclear to what extent these mechanisms are relevant for the 8–17 DNAzyme catalysis in the presence of other divalent metal ions cofactors, more important for activity in vivo, due to the global structural change that the latter impose on the DNAzyme. Some similarities are already known to be present, e.g., **G14** was shown[15] to act as a general base also in the presence of $Mg^{2+}$, but no complete mechanistic model can be proposed without a high resolution structural information on the DNAzyme fold that these ions induce.

In the current study we set out to solve the solution structure of the 8–17 DNAzyme in the presence of $Zn^{2+}$ and to characterize its interactions with this ion. A shortened, unimolecular construct of 8–17 DNAzyme (Fig. 1C) facilitating NMR structural studies is designed,

validated for unaltered catalytic activity, and characterized by NMR. Most surprisingly, while pronounced spectral changes are indeed observed upon $Zn^{2+}$ titration, confirming ion-induced folding, the high-resolution structure that we solve is very similar from the previous crystallographic ones in terms of both the fold of the catalytic domain and the positioning of substrate-binding arms.

To understand this unexpected result, we confirm by NMR that moderately high (0.2–0.4 M) concentrations of $Na^+$ induce the same tightly folded structure as $Zn^{2+}$, as was already suggested previously[11]. Given significant $Na^+$ concentrations present in the crystallization conditions[13], we propose that the previous crystal structures captured the highly compacted structure usually associated with ions such as $Zn^{2+}$ or $Mg^{2+}$ instead of the hypothesized open apo form (Fig. 1D). Moreover, most unexpectedly, NMR experiments performed in presence of $Pb^{2+}$ reveal that even this cofactor induced 8–17 DNAzyme global folding at NMR sample concentrations (>1 mM). Based on these results, reinforced by a series of activity assays and comparisons with previous literature we bring forward and discuss the idea that $Pb^{2+}$ might carry out its catalytic role in the same folded state as other ions, even if at most often employed $Pb^{2+}$ concentrations the population of this state might remain low.

Moreover, this study brings important details regarding divalent ion binding by the 8–17 DNAzyme with the identification of the $Zn^{2+}$ binding site responsible for the ion-induced folding. However, given the position of this site away from the scissile phosphate, we conclude that it has to be of purely structural importance and another $Zn^{2+}$ ion must be required for catalysis. While our current attempts at capturing also this second catalytic $Zn^{2+}$ site were not successful, the results presented herein establish a much more intricate picture of 8–17 DNAzyme's interactions with divalent ions with distinct structural and catalytic metal-binding sites being involved, similarly to what was previously proposed for several naturally occurring ribozymes[24–28].

## Results

### Construct optimization and validation

To facilitate a high-resolution NMR study, an 8–17 DNAzyme construct of the shortest length is highly desired. While the catalytic core of 8–17 does not tolerate the deletion of any nucleotides, with the exception of **N15.0**[8] (Fig. 1A), the substrate binding arms, usually containing over a dozen base pairs each, can potentially be significantly shortened without disrupting the DNAzyme's structure. The reduced propensity for hybridization between the DNAzyme and substrate strands can be counteracted by connecting the two into a single unimolecular construct, preferably with the ultra-stable GAA triloop[29,30].

Following this general template, a series of 8–17 constructs with different arm sequences were designed and evaluated by NMR in the presence of $Zn^{2+}$, as described in detail in the Supplementary Note 1 and Supplementary Fig. 1. Throughout this optimization the catalytic domain was kept unaltered with respect to the crystallographic study to facilitate a structural comparison, with the exception that **A15.0** was deleted. As is a common approach in structural studies of DNAzymes the tested constructs were synthetized in all-DNA form to abolish self-cleavage and only the selected variant, presented in Fig. 1C and referred to as 8–17_short from now on, was also obtained in a cleavable form and tested for catalytic activity. Cleavage rate constants of 0.453 min$^{-1}$, 0.728 min$^{-1}$ and 0.063 min$^{-1}$ were measured (Supplementary Fig. 2a) in the presence of 700 μM $Zn^{2+}$, 50 μM $Pb^{2+}$ and 50 mM $Mg^{2+}$, respectively (at pH 6.0). In the same conditions the rate constants measured for bimolecular full-length 8-17 construct used in the crystallographic study[13] (Supplementary Fig. 2b) were 0.246 min$^{-1}$, 0.488 min$^{-1}$ and 0.100 min$^{-1}$, showing that 8–17_short retained catalytic capabilities comparable to longer, bimolecular DNAzyme variants. The activities of the two constructs were also studied as functions of the three divalent metal ion concentrations (Supplementary Fig. 3).

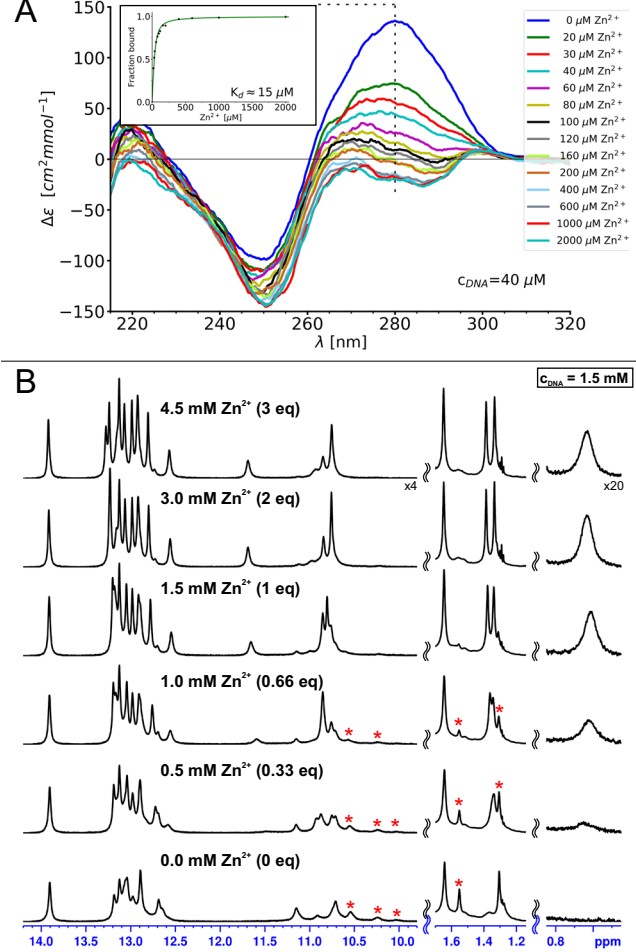

**Fig. 2 | Zn²⁺-induced folding of 8–17_short. A** monitored by CD spectroscopy and (**B**) monitored by 1D ¹H-NMR. NMR resonances belonging to an initially present second spectral form are marked by red stars. The titration was performed in 10 mM sodium cacodylate buffer which originally contained no additional salts. Source data are provided as a Source Data file.

The shortened construct was tested for Zn²⁺-induced folding by NMR and CD spectroscopies. The results of the CD-monitored titration with Zn²⁺ are presented in Fig. 2A. They show gradual and saturable spectral changes in response to the metal ion. The effect is much more pronounced here than originally reported[11], probably due to a smaller B-helical contribution from the DNAzyme's arms. The spectral changes in function of Zn²⁺ concentration can be fitted to a 1:1 interaction model, yielding an apparent $K_d$ value of ≈15 μM (previous literature reported 52 μM[9], 18.9 μM[10] and 3.1 μM[11] for different full-length 8–17 constructs). Given such an apparent $K_d$, the binding of Zn²⁺ to 8–17_short should be practically quantitative at NMR sample concentrations (>1 mM). A Zn²⁺ titration monitored by 1D ¹H-NMR presented in Fig. 2B confirms this conclusion. At least two spectral forms can be observed in the imino and methyl regions of the spectrum when no Zn²⁺ is present. As the first equivalent of Zn²⁺ is titrated many of the peaks gradually disappear (marked with red stars in Fig. 2), until only one spectral form remains. The folding is also accompanied by the gradual appearance and sharpening of new resonances, like the one observed at around 0.6 ppm in Fig. 2B. Additional equivalents of Zn²⁺ give rise only to subtle movements of some peaks, indicating that the folding transition is already saturated. Similar results were obtained when experiments were repeated at pH 7 (Supplementary Fig. 4), confirming that the lower pH used did not perturb the structural transition. Overall, our shortened construct turned out to behave very

similarly to longer 8–17 variants in terms of both the catalytic activity and Zn²⁺-induced folding. From now on, when referring to specific residues in 8–17_short, we will use the numbering specific to this construct (Fig. 1C) adding whenever relevant the standard numbering of the residues of the catalytic domain (Fig. 1A) in parenthesis in bold e.g., G29(**G14**).

## NMR structure of 8–17_short in presence of Zn²⁺ is similar to Pb²⁺-bound crystal structure

While one equivalent of Zn²⁺ is enough to saturate the folding transition at NMR concentrations, the folded structure is not particularly thermally stable in these conditions. For this reason, for NMR structure determination a sample containing 2 eq. of Zn²⁺ as well as 200 mM of NaCl to increase the ionic strength was used, for which good quality spectra could be recorded up to 35 °C. NMR spectral assignments were obtained using standard approaches (see Methods) and confirmed through isotope labeling of key residues within the catalytic domain (Supplementary Fig. 5). A detailed description of observed structurally relevant NMR spectral features, including the set of measured long-range NOEs, can be found in the Supplementary Note 2 complemented by Supplementary Figs. 6 and 7. The NMR structure of 8–17_short in the presence of Zn²⁺ was determined through a restrained molecular dynamics protocol described in Methods, with the structure determination statistics provided in Supplementary Table 1. The fold of the catalytic domain is very well defined in the NMR structural bundle (RMSD = 0.99 Å), with a bit more uncertainty regarding the mutual positioning of the substrate-binding arms, resulting in a RMSD of 1.83 Å for the entire structure.

Apart from the three helices presented in Fig. 1A, the structure (Fig. 3; the whole NMR ensemble presented in Fig. 4) also features the G22(**G7**)-C28(**C13**), A21(**A6**)-G29(**G14**) and G6(**rN18**)-A30(**N15**) base pairs (Fig. 3B), that were previously only observed in the crystal structures of the apo and Pb²⁺-bound forms of the enzyme. These base pairs are crucial for the formation of the L-platform[23] tertiary fold in the catalytic domain and for positioning G29(**G14**) base to act as a general base during catalysis. The G22-C28 pairing is directly visible in the NMR spectra through strong NOE contacts between the exchangeable protons of the two bases. On the other hand, sheared G:A base pairs rely on hydrogen bonding between groups that are difficult to directly observe by NMR (amine groups of G and A residues), but their presence is required to satisfy a set of long range NOE contacts between nucleobase and sugar protons of the residues involved (listed in Supplementary Table 2). The tertiary fold of the DNAzyme is defined by the co-axial stacking of helices 1–4 and 2–3 (Fig. 3A) which fixes the entire structure into a V-like shape with the interhelical angle between the two substrate binding arms averaging 83° ± 11° for different conformers within the NMR bundle.

Overall, the catalytic domain fold that we observe in the presence of Zn²⁺ is very similar to the one reported previously in the crystal structures, with the average catalytic domain RMSD of the members of our NMR bundle to the crystal structure equal to 1.28 Å (which is just beyond the uncertainty of the bundle itself). When the phosphate group of T27(**N12**), the only moiety whose position in the crystal structure appears to fall outside the range of uncertainty of the NMR bundle upon visual inspection is omitted the RMSD further drops to 1.18 Å. Also, the interhelical angle between substrate-binding arms encountered in the crystal (≈75°) is close to what we observe in our NMR structure. The degree of similarity between the two structures can be appreciated in Fig. 4.

## Mg²⁺ and Na⁺ induce the same overall 8–17_short fold as Zn²⁺

Previous literature suggests that other divalent ions, such as Mg²⁺ as well as high concentrations of monovalent ions, like Na⁺, induce a folding transition in the 8–17 DNAzyme, akin to that promoted by Zn²⁺[9–11]. We decided to verify whether such behavior can also be

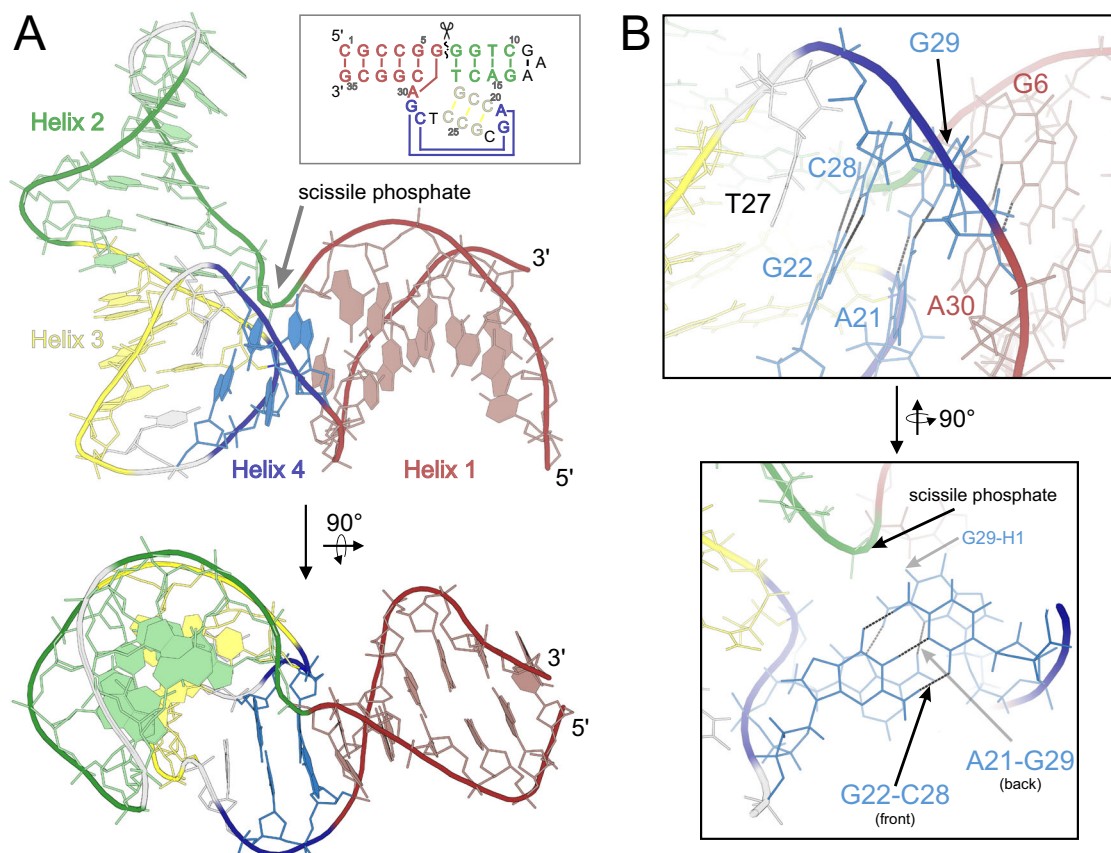

**Fig. 3 | The solution structure of 8-17 DNAzyme in the presence of Zn²⁺. A** the overall fold, (**B**) geometric details of Helix 4.

observed for our 8−17_short construct. We have started the investigation by performing CD- and NMR-monitored titrations of 8−17_short with MgCl₂ and NaCl, presented in Fig. 5. The induced spectral changes are indeed qualitatively similar to what was observed for Zn²⁺ (Fig. 2), yet require several equivalents of Mg²⁺ to saturate even at NMR concentrations (Fig. 5B; apparent $K_d \approx 90$ μM from CD measurements) and even orders of magnitude more metal ion is needed in case of Na⁺ (Fig. 5A; apparent $K_d$ of Na⁺ ≈90 mM), consistent with the literature[9–11]. To confirm that tertiary folds induced by these ions and Zn²⁺ are indeed the same, we have performed NMR resonance assignments and analyzed the NOE patterns for 8−17_short in the presence of either 7.5 mM Mg²⁺ or 400 mM NaCl. While the lower spectral quality made accurate integration of many peaks difficult and thus precluded structure calculation in these conditions, the observed patterns of long-range NOE connectivities and characteristic chemical shifts was qualitatively the same as for Zn²⁺ (as described in detail in Supplementary Note 3 and illustrated in Supplementary Fig. 8). One can thus conclude that Mg²⁺ and Na⁺ indeed induce the same overall DNAzyme fold as Zn²⁺. To retroactively exclude the possibility that the structure determined for sample containing both Zn²⁺ and 200 mM Na⁺ was dictated by the monovalent ion rather than by Zn²⁺, we have also performed the 2D-NMR analysis for a sample containing only Zn²⁺ without added NaCl. Once again, the same pattern of long-range NOEs was observed (see Supplementary Note 3, Supplementary Table 2 and Supplementary Fig. 8), confirming that one equivalent of Zn²⁺ alone is indeed enough to induce the folded structure.

**Even Pb²⁺ ions induce global folding at NMR concentrations**

As control experiments we have also performed NMR and CD titrations with Pb²⁺, the only activating ion not shown to induce 8−17 DNAzyme folding[9–11]. Surprisingly even in this case we observed extensive spectral changes in both techniques. The CD titration (Fig. 6A) initially

followed a pattern similar to what was observed for the other ions (Figs. 2 and 5) with a gradual decrease of the band around 280 nm. Yet, starting at around 2 equivalents Pb²⁺ and before the initial structural transition could be saturated, a new spectral band started to appear centered at around 320 nm. This second transition saturated at around 10–15 equivalents Pb²⁺. Similarly, in the first part of the NMR titration (Fig. 6B) we have once again observed a transition from multiple spectral forms to only one, alike of what was described above for Zn²⁺, Mg²⁺ and Na⁺. Unlike the CD titration this transition saturated at around 2 eqivalents of Pb²⁺, due to much higher concentrations of interaction partners used in NMR, as demonstrated by the virtual disappearance of the peaks marked with red stars in Fig. 6B. The addition of the additional equivalents of Pb²⁺ finally resulted in the appearance of a small fraction of a distinct structural form (with a distinct set of imino proton resonances located around 12 ppm; green stars in Fig. 6B), probably equivalent to the one responsible for the 320 nm band in CD. To confirm this assumption we have performed another NMR titration (Fig. 6C) at a DNA concentration more akin to what we used for CD spectroscopy. In this case when 10 equivalents of Pb²⁺ was reached the structural form with imino resonances around 12 ppm (green stars in Fig. 6C) became the dominant one, in accordance with the CD data. Thus overall, the addition of Pb²⁺ ions to 8−17_short causes two consecutive structural transitions: (1) one from the initial ensemble of structures to a single folded state that becomes the only form present at around 2−3 equivalents of Pb²⁺ at NMR concentrations (Fig. 6B) and (2) a second transition to a different folded state at higher Pb²⁺-to-DNA molar ratios. This second transition is accompanied by the disappearance of imino proton resonances of Watson-Crick base-pairs (Fig. 6C), including those from the DNAzyme's arms, and thus we would like to argue that it is a reflection of the known tendency of Pb²⁺ ions to destabilize helical folds at higher molar ratios by direct coordination of nucleobases, especially

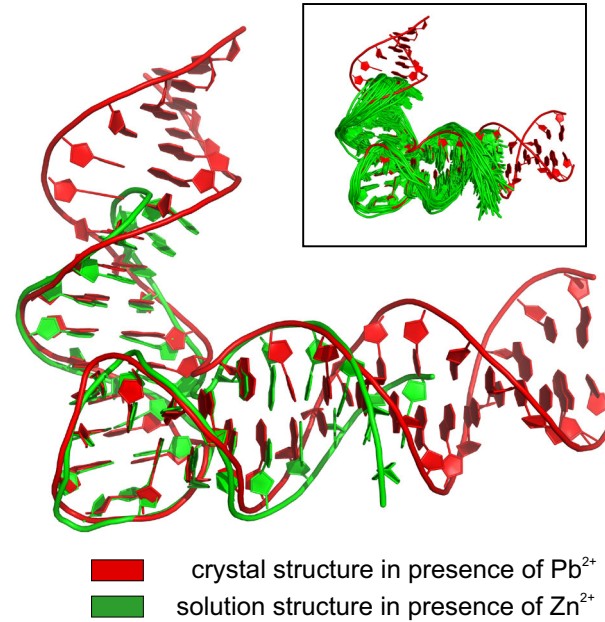

**Fig. 4 | Comparison between Pb²⁺ and Zn²⁺ stabilized structures.** Main panel uses a single conformer from the NMR bundle for clarity, while the insert shows the entire bundle.

crystal structure in presence of Pb²⁺

solution structure in presence of Zn²⁺

guanines[31]. The rather early onset of this transition for 8–17_short may be related to its very short and G-rich helices. This argument is presented in detail in the Supplementary Discussion 1 (strengthened by additional experimental data presented in Supplementary Figs. 9 and 10), culminating in a suggestion that the structure formed is a Pb²⁺-stabilized G-quadruplex. Here we will instead focus on the fold resulting from the initial transition. A set of 2D NMR spectra was recorded and analyzed for a sample containing 2.33 equivalents of Pb²⁺ for which only a single structural form was present. Similarly to the previous samples containing only a single type of ion (Na⁺, Mg²⁺ and Zn²⁺) some of the NMR resonances were broadened making the accurate integration of their NOESY cross-peaks difficult. Still a fairly complete resonance assignment was possible (Supplementary Data 1 and 2) and a significant number of long-range NOESY connectivities was observed within the catalytic domain (Supplementary Table 2). Both these long-range NOE contacts and non-standard chemical shifts were very similar to the ones observed in all the samples discussed thus far, as described in detail in the Supplementary Note 3, which is only possible if the overall fold of the molecules is conserved throughout the studied conditions. Thus, at least for our 8–17_short construct, Pb²⁺ appear to induce the same overall 8–17 DNAzyme fold as other metal ions at NMR concentrations.

### Control experiments with full-length 8-17 DNAzyme

As the just described folding of the 8–17_short DNAzyme by Pb²⁺ ions is at odds with the current state of knowledge about this system, we have repeated the CD and NMR-monitored titrations with all four metal ions for a more standard 8–17 DNAzyme variant – the bimolecular, full-length construct used previously for the crystallographic study. Surprisingly, also for this construct our CD titrations revealed qualitatively similar spectral changes for all four metal ions – a saturable decrease of intensity of the CD band around 280 nm, although with somewhat different end-points for the different metal ions (Fig. 7). This discrepancy with the previous results[11] cannot be attributed to difference in experimental conditions, but is rather the effect of specific DNAzyme sequences being studied, as when we repeated the CD experiments using DNAzyme variants used in reference 11 we obtained the same results as the original authors (only marginal CD spectral changes

upon Pb²⁺ titration, Supplementary Fig. 11). We are currently conducting a more systematic study using different 8–17 variants to shed more light on this issue in the future, yet the results presented in Fig. 7 show that at least the crystallographic 8–17 DNAzyme construct undergoes structural changes upon Pb²⁺ titration. The NMR titrations (Supplementary Fig. 12) show less spectacular spectral changes due to significant spectral crowding in 1D NMR severely obstructing the observations, nevertheless the changes that can be observed are very similar among all studied metal ions – gradual shifts of the same two methyl proton resonances as well as the appearance of a new imino resonance around 13.1–13.2 ppm and a highly shielded proton resonance around 0.9 ppm (Supplementary Fig. 12).

### Are the global folds assumed by 8–17 DNAzyme in presence of Zn²⁺ and Pb²⁺ the same?

It is a long-standing literature consensus that Pb²⁺ binds to the open, apo form of 8–17 DNAzyme without affecting its original conformation, while Zn²⁺ binding triggers a global structural rearrangement towards a more compact fold (Fig. 1D). Despite this, the solution structure of Zn²⁺-bound 8–17 DNAzyme that we report here is practically indistinguishable from the Pb²⁺-bound and apo crystal structures, in terms of catalytic domain fold and substrate-binding arms orientation (Fig. 4). Clearly, for some reason, all these structural studies have captured the same state of the enzyme, yet which one is it – the apo one or the ion-induced one? We wish to argue that it is the latter given that our NMR and CD titrations clearly indicate that the structure we solved was only formed upon Zn²⁺ addition (Fig. 2). The appearance of such ion-induced fold in crystals formed in presence of Pb²⁺ and even in those without any divalent ions would of course be highly unexpected, yet we believe that the bulk of experimental results presented in the current work allows us to propose a way to reconcile these results.

Regarding the apo state, our 1D NMR spectra recorded before addition of any metal ions (e.g., in Fig. 2B, bottom) show the enzyme to be present as a mixture of different conformations, which then gradually collapse into a single state upon metal ion titration. The complexity of such mixture precludes any detailed structural characterization of the different states forming it, however, as some monovalent ions are always present to compensate the negative charge of DNA, this conformational ensemble must contain a fractional population of the folded state (Fig. 8, top). When a biomolecule is dynamic/polydisperse, crystallization often selects only a single among the forms present to construct the crystal lattice, usually the more structured, rigid one. We thus find it plausible that the tightly folded, ion-induced structure was selected from the conformational ensemble of 8–17 DNAzyme during crystallizations. To strengthen this argument, one can notice that monovalent ions that also promote folding were present in crystallization drops, in quantities which could make the folded state abundant. The protein AsfvPolX, crystalized in complex with 8–17 DNAzyme, was supplied in 300 mM NaCl[13] (final concentration not provided) and two out of three crystallization conditions also contained 200 mM NH₄⁺. The presence of what we argue to be the compact form of the DNAzyme in the Pb²⁺-containing crystals could also be rationalized using the same argument (conformational selection when forming the crystal lattice). However, on top of that our Pb²⁺ titrations (Figs. 6 and 7) show that Pb²⁺ ions themselves can at higher concentrations – as used for both NMR and crystallographic structure determination studies – induce the compact fold of the DNAzyme.

This finding itself may be the most unexpected result of our investigation, as previous FRET and CD studies have shown no signs of Pb²⁺-induced folding. These studies were of course performed at different DNAzyme concentrations, using different sequence variants and sample conditions, including much higher metal-to-DNA molar ratios. In the Supplementary Discussion 2 we discuss how these differences

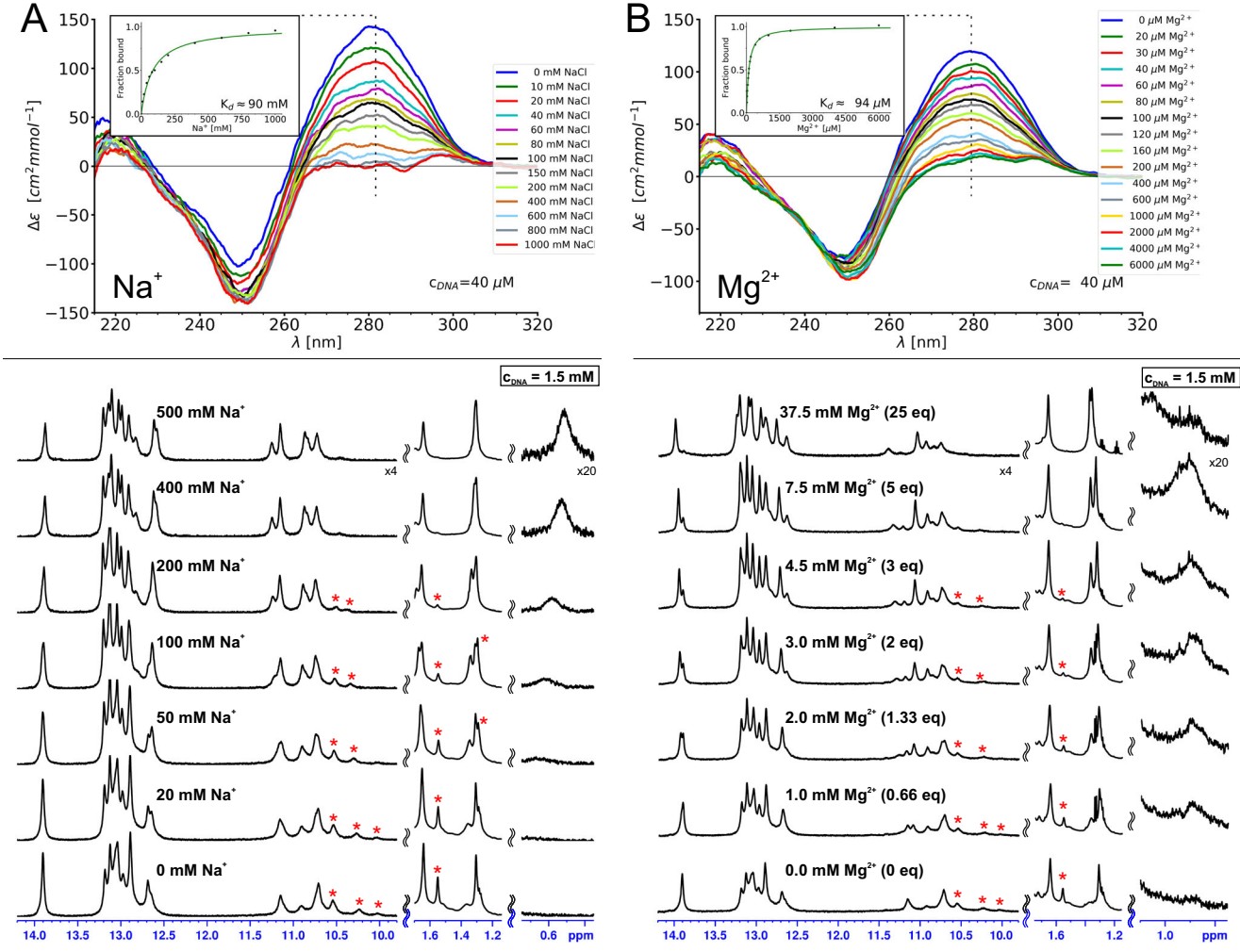

**Fig. 5 | Na⁺- and Mg²⁺-induced folding of 8-17_short. A** titrations with Na⁺ (**B**) titrations with Mg²⁺. For each ion the folding was monitored by CD spectroscopy (top) and 1D ¹H-NMR (bottom). NMR resonances belonging to an initially present second spectral form are marked by red stars. The titration was performed in 10 mM sodium cacodylate buffer which originally contained no additional salts. Source data are provided as a Source Data file.

might have prevented the observation of Pb²⁺-induced folding. Here, we would like to bring forward one significant difference between Pb²⁺ and other activating ions that is apparent in our dataset and that could explain why Pb²⁺-induced folding was not considered in previous studies of the DNAzyme. Namely, in case of both Zn²⁺ and Mg²⁺ we observe a several orders of magnitude wide discrepancy between the metal ion concentrations needed to induce DNAzyme folding and those needed to saturate its catalytic activity (Table 1). Concentrating on Zn²⁺, for the 8–17_short the apparent $K_d$ for Zn²⁺ binding estimated from the CD-titration is 15 μM, while the $K_d$ derived from catalytic activity measurements is instead around 4 mM (Supplementary Figure 3). Similarly mismatched $K_d$ values were also obtained for the full-length construct: 23 μM based on CD data and 3 mM based on activity assays (Table 1). Indeed, similar discrepancies are present throughout the available literature between the Zn²⁺ affinities estimated using 8–17 DNAzyme folding (always micromolar)[9–11] and those derived from the Zn²⁺ concentration dependence of the measured rate constants $k_{obs}$ (always millimolar)[9,32].

This conundrum may suggest that at least two Zn²⁺ (or Mg²⁺) ions are involved in specific interactions with 8–17, one causing its folding (binding with micromolar $K_d$ values) and another one activating the DNAzyme for cleavage (and having a millimolar affinity). A similar conclusion was already suggested in the first paper reporting ion-induced folding of 8–17 DNAzyme[9], yet for some reason did not find

much reflection in later literature. The affinity discrepancy between these two sites leads to a situation in which at Zn²⁺ concentrations when cleavage rates become appreciable (hundreds of μM, Supplementary Fig. 3) the DNAzyme is already fully folded (Fig. 8, right branch).

On the other hand for lead ions the dissociation constants estimated using CD and activity assays appear to be much more similar (the CD-derived values cannot be compared to previous literature, as Pb²⁺-induced folding has not been reported before, while the activity-derived ones are consistent with previous reports within order of magnitude[9]). In such a case the Pb²⁺ induced activity could already be high at metal concentrations like dozens of μM for which the DNAzyme is only partially folded into the active state (Fig. 8, left branch). Even the presence of only a fraction of the active folded state at a given time can give rise to the high final cleavage yields observed at these Pb²⁺ concentrations, because as the active state is gradually consumed and ions are released the dynamic equilibrium between the different DNAzyme conformations would work to repopulate it.

Thus, in our opinion it may be reasonable to argue that the explanation for the unexpected coincidence of the Zn²⁺-induced solution structure and the Pb²⁺-bound crystal one is that the two ions actually use the same folded state for catalysis. The main difference between the two ions would be the extent to which the active state is populated at metal concentrations capable of inducing appreciable

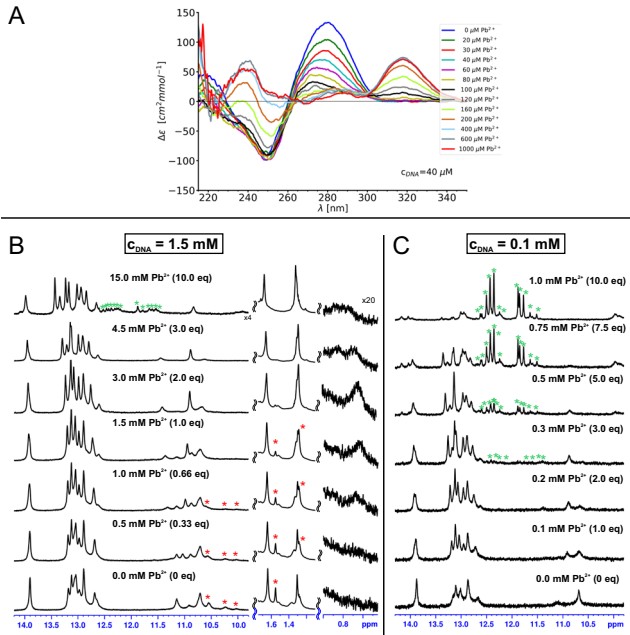

**Fig. 6 | Pb²⁺-induced folding of 8-17_short. (A)** monitored by CD spectroscopy and **(B)** monitored by 1D ¹H-NMR at millimolar DNA concentrations, **(C)** monitored by 1D ¹H-NMR at submilimolar DNA concentrations. NMR resonances belonging to an initially present second spectral form are marked by red stars while those to the form appearing at higher concentrations of Pb²⁺ with green ones. The titration was performed in 10 mM sodium cacodylate buffer which originally contained no additional salts. Source data are provided as a Source Data file.

catalytic activities − the Zn²⁺-induced activity occurs only at concentrations at which the DNAzyme is already fully folded, while Pb²⁺ starts to activate the DNAzyme even without significantly increasing the population of the folded state in the conformational ensemble of the apo state.

Formulated in terms of metal-DNA interactions, this hypothesis would argue for the existence of two distinct structural and catalytic binding sites. Ions such as Zn²⁺ and Mg²⁺ would have a much higher affinity of the structural site and thus saturate it preferentially, only starting to occupy the catalytic site at higher molar excesses (Fig. 8, right branch). Pb²⁺ on the other hand would either have the two affinities much closely aligned (Fig. 8, left branch) or use only a single binding site to achieve both DNAzyme activation and folding.

While this proposal provides a different interpretation of the available ion-induced folding data[9–11] to the one currently accepted in the literature – a single active conformation populated to a different extent with Pb²⁺ compared to other ions, in place of the postulate of two different active states –, we believe that it is not directly at odds with previous experimental observations (see Supplementary Discussion 2 for a more thorough discussion).

If our hypothesis was true, then it should be possible to enhance Pb²⁺-induced activity at its typical concentrations (dozens-to-hundreds μM) by the addition of other metal ions − like Mg²⁺ − in quantities that are not enough to induce appreciable activity themselves, but are capable of significantly increasing the population of the folded active state for Pb²⁺ to use. On the other hand, if Mg²⁺ and Pb²⁺ use different global folds of the DNAzyme for catalysis then similar Mg²⁺ additions should inhibit the Pb²⁺-induced activity by transforming the DNAzyme into a fold that Pb²⁺ cannot use. Thus, to test our structural proposal we have measured the cleavage rates for two variants of 8−17 DNAzyme in presence of 50 μM of Pb²⁺ alone and alongside increasing additions of Mg²⁺, up to 500 μM (these concentrations of Mg²⁺ by themselves induce very little cleavage even after 2 h). For both variants we have

indeed observed a gradual increase in activity alongside increased Mg²⁺ doping (Supplementary Fig. 13), as our one active state hypothesis would predict. Moreover, a similar observation was recently published regarding the effect of Na⁺ ions on 8-17 DNAzyme catalytic activity in the presence of Pb²⁺[18]. However, it has to be noted that the authors of that paper assigned the observed effect to a direct involvement of Na⁺ ions in Pb²⁺-induced cleavage reaction proposed by a previous MD study[19], rather than to a structural shift.

## Identification of the structural Zn²⁺ binding site
In the NMR experiments metal ions such as Zn²⁺ can be only observed indirectly through their influence on chemical shifts of the atoms within the macromolecule they bind to (chemical shift perturbations – CSPs). If the metal ion binds without affecting the overall structure of the macromolecule then the localization of CSP carry information about where the binding occurs. The possibility of inducing the folded state of 8−17_short using an ion with orders of magnitude weaker affinity (Na⁺) opens the possibility of studying Zn²⁺ binding using this principle. When Zn²⁺ is titrated to the Na⁺-stabilized DNAzyme (Fig. 9) it displaces the monovalent ion from the binding site(s), without changing the global structure, only affecting the local chemical environment of atoms close to the binding site. As seen in Fig. 9a, b, among the imino and aromatic protons the highest CSPs are recorded for the residues G7(**N1**) and T17(**T2**) forming the G:T base pair, adjacent to the cleavage site, suggesting Zn²⁺ interaction at this site. When all the measured CSPs (base, sugar, phosphate) are visualized on the structure (Fig. 9c) the shifts adjacent to the G:T base pair remain dominant, yet some moderate CSPs can be seen all throughout the catalytic domain. Their appearance could be related either to some local structural differences between the Zn²⁺ and Na⁺-stabilized structures or to the fact that with Na⁺ alone the population of the folded structure was not 100%. Interestingly, only a relatively small CSP of 0.09 ppm was measured for the ³¹P atom of the scissile phosphate, suggesting no direct Zn²⁺ interaction with this group.

Subsequent titration of two additional equivalents of Zn²⁺ yielded much smaller CSPs in the vicinity of the G:T base pair (Fig. 9A, B), confirming near-saturation of this binding site after just one equivalent. Instead, two regions of moderate CSPs appear, probably corresponding to additional weaker interaction sites (CSPs visualized on the structure in Supplementary Fig. 14b). One of them was located in the vicinity of residues C23(**Y8**) and G24(**N9**) (Helix 3 and its capping residue) and the other in one of the substrate-binding arms. The location of these additional sites, on the far peripheries of the molecule, makes any involvement of the metal bound there in the catalytic process highly unlikely. Similar titrations were also performed for the sample already structured by one equivalent of Zn²⁺, yet containing no added NaCl. In this case again, additional equivalents of Zn²⁺ induced some moderate CSPs around the two above-mentioned weak binding sites, but not around the G:T base pair (Supplementary Fig. 14c). This confirms that in these conditions this binding site is already saturated and thus it is the one responsible for the Zn²⁺-induced folding (the structural site introduced above).

While the CSPs themselves cannot reveal the exact geometry of this interaction, their pattern suggests that Zn²⁺ is interacting with the Hoogsteen edge of the G7(**N1**) residue which forms the G:T base pair. First, both the imino and aromatic protons of G7(**N1**) experience strong CSP, while for the thymidine only the imino proton is affected, with almost no CSP on the H6 and methyl protons (Fig. 9C). Second, the only other aromatic proton with a strong CSP belongs to G8, which directly stacks with G7(**N1**), further locating the binding site on this strand of the helix. Given that guanosine's Hoogsteen edge is the strongest interaction site for Zn²⁺ on single nucleotide level[33], these CSP data are most readily explained by the binding to

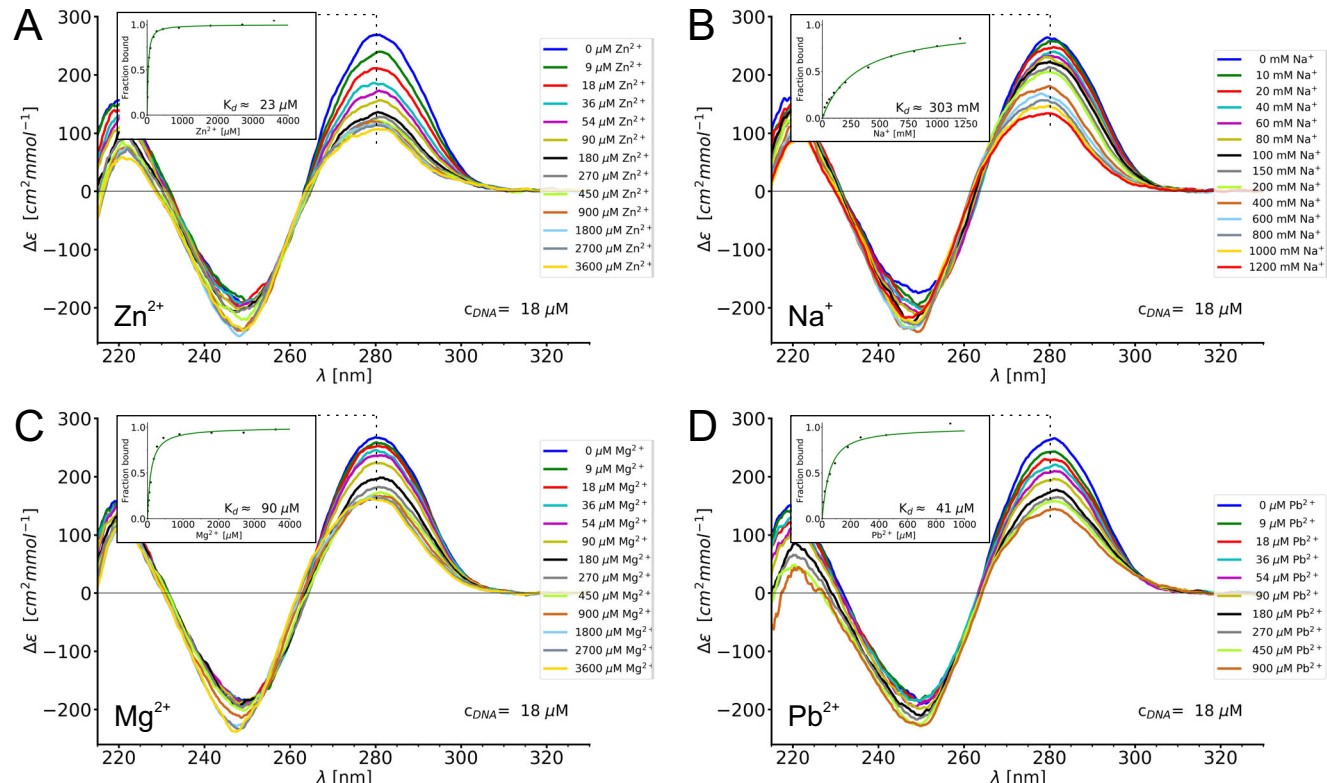

**Fig. 7 | Ion induced folding of the 8-17 DNAzyme construct used in the previous crystallographic investigation (sequence in Fig. 1B).** CD-monitored titrations with four different metal ions: (**A**) $Zn^{2+}$, (**B**) $Na^+$, (**C**) $Mg^{2+}$ and (**D**) $Pb^{2+}$. Source data are provided as a Source Data file.

G7(**N1**). As to why $Zn^{2+}$ interaction with such a site could facilitate the DNAzyme's folding, one can notice that in our structure the Hoogsteen edge of G7 directly faces the phosphate group of T27(**N12**) (the N7 atom of G7 is located 5.5–8.0 Å from the phosphorus atom in our structural bundle). Electrostatic repulsion between these two entities might represent a major obstacle for folding that is removed when a divalent cation binds and shields this repulsive interaction. The influence of the $Zn^{2+}$ ion on the local environment of T27(**N12**) phosphate is reflected in the 0.25 ppm $^{31}P$ CSP, the highest among all phosphate groups in the molecule. The involvement of G7(**N1**)-N7 and T27(**N12**)-P in $Zn^{2+}$ binding was tested by studying the folding of 8–17_short variants containing either the 7-deaza or phosphorothioate modifications at the respective sites (see Supplementary Fig. 15 and Supplementary Note 4). For the 7-deaza-G7(**N1**) construct a fivefold decrease in binding affinity was observed, while for the T27(**N12**) phosphorothioate modification the decrease was two- to fourfold, depending on the method of data fitting (see Supplementary Note 4). These $K_d$ changes confirm the involvement of these two moieties in the interactions, yet their magnitudes may point towards second sphere interactions, rather than direct metal coordination. On the other hand, the $Zn^{2+}$ ion bound at this site almost certainly does not directly interact with the scissile phosphate, as (1) the $^{31}P$ CSP of this group is small throughout our titrations and (2) the phosphorothioate substitution at this site does not influence observed ion binding (Supplementary Fig. 15). Available literature data provides strong evidence that the catalytically-required $Zn^{2+}$ ion should have a direct interaction with the scissile phosphate, through its $R_p$ oxygen atom[17]. This reinforces the idea that the strong site we observe is not the catalytic one. The lack of CSPs in the direct vicinity of the scissile phosphate indicate that this catalytic site was not appreciably populated in our titrations, even though $Zn^{2+}$ concentrations of up to 4.5 mM were reached (Fig. 9 and S13). This might be either due to other competing sites of similar affinity (a factor which does not play such an important role during kinetic experiments performed at much lower DNA concentrations and thus many-fold excess of the metal) or to some distortion of the catalytic site in the all-DNA construct used for structural studies (the lack of the 2'OH at residue G6(**rN18**) might alter both the local conformation and electrostatic potential).

## Discussion

The crystal structures of the 8–17 DNAzyme[13] provided much momentum for the in-depth studies of its catalytic mechanism, as described in the Introduction. However, due to the consensus that the available structures represented the open/apo state of the enzyme (Fig. 1B), all the conclusions of these studies were believed to only be directly applicable to $Pb^{2+}$-driven catalysis. With our current finding that the DNAzyme fold induced by $Zn^{2+}$ and other ions is the same as one found in the crystals and follows the L-platform[23] architecture, many of these conclusions can be extended to catalysis in the presence of other divalent metal ion cofactors.

First, the V-like overall fold of the DNAzyme, believed to orient the 2'OH group of r**N18** (Fig. 2B) for in-line attack on the scissile phosphate (α-catalysis), is retained in the $Zn^{2+}$-stabilized structure suggesting that this mechanism of catalysis is relevant in presence of all divalent metal ions. Second, with **G14** (G29 in our construct) nucleobase sharing its position in both the crystal and solution structures, its role as general base appears to also be conserved regardless of the metal ion present. This conclusion is supported by the previous literature, as the role of **G14** as the general base was already proven experimentally also in the presence of $Mg^{2+}$[15]. On the other hand, much less can be inferred about the structural aspects of catalytic involvement of divalent ions other that $Pb^{2+}$, as our current study did not manage to locate the weak ($K_d > 1\,mM$)[9] catalytically relevant $Zn^{2+}$ binding site. Given the same DNAzyme structure, this interaction site might actually be similar to the one occupied by $Pb^{2+}$

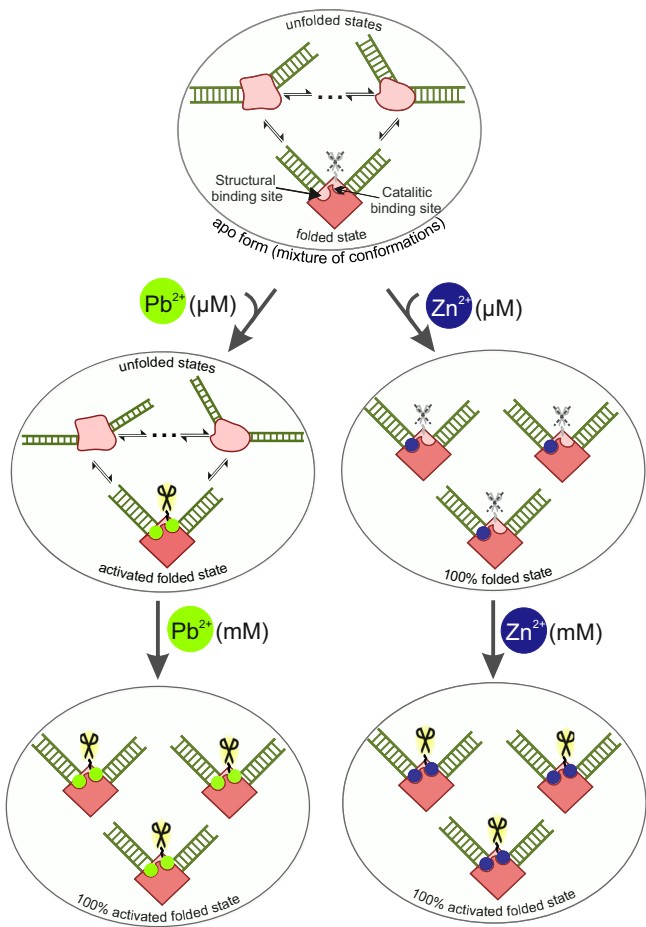

**Fig. 8 | The proposed model for the interplay between divalent metal ions and 8–17 DNAzyme structure and activity.** The top part of the scheme represents the conformational ensemble of the apo- form of the enzyme, while the effects of $Pb^{2+}$ and $Zn^{2+}$ ions are depicted below in the left and right panels, respectively. The occupation state of structural and catalytic metal binding sites is also shown in each case for the well-folded DNAzyme molecules. When the catalytic site is filled the DNAzyme is activated, as marked by the black, highlighted scissors.

**Table 1 | Apparent dissociation constant ($K_d$) values for the interaction of 8–17 DNAzyme variants with different divalent ions derived from global folding and catalytic activity measurements**

| Metal ion | DNAzyme construct | $K_d$ of metal:DNA interaction estimated using CD spectroscopy | $K_d$ of metal:DNA interaction estimated using activity assays |
|---|---|---|---|
| $Zn^{2+}$ | 8–17_short | 15 μM | 4 mM |
| | crystallographic | 23 μM | 3 mM |
| $Mg^{2+}$ | 8–17_short | 94 μM | 14 mM |
| | crystallographic | 90 μM | 9 mM |
| $Pb^{2+}$ | 8–17_short | <100 μM[a] | ~0.1 mM |
| | crystallographic | 41 μM | ~0.2 mM |

[a]$Pb^{2+}$ affinity to 8–17_short is impossible to accurately derive from the CD titration, as a second structural transition commences before the first one is saturated (see text). The estimate given in the table is based on the first few datapoints in CD titration, assuming titration endpoint being $\Delta\varepsilon_{280nm}\approx0$, similar to other metals.

in the crystal, however, with somewhat different interaction geometry to allow for the direct coordination of the scissile phosphate[17]. Interestingly, MD simulations preformed previously for the $Pb^{2+}$-bound crystal structure featured a first sphere contact with $R_p$ oxygen even for this metal[19], indicating that such a geometry is at least electrostatically favorable. Establishing whether other activating metal ions utilize such a binding mode requires further experimental or computational investigations. Nevertheless, the structural data presented here sets the stage for a more in-depth mechanistic characterization of 8–17 DNAzyme catalysis in the presence of non-$Pb^{2+}$ cofactors, such as $Mg^{2+}$, and eventual design of its more in vivo active variants.

## Methods
### Sample preparation
All the all-DNA constructs (including the 7-deaza-dG and phosphorothioate modified ones), as well as their cleavable, fluorescein labeled DNA/RNA hybrid counterparts for kinetic experiments were purchased from Metabion GmbH. The site-specifically $^{13}C/^{15}N$ labeled samples were synthesized in house using phosphoramidites purchased from Silantes GmbH. Before usage, the all-DNA constructs were purified from residual organic cation detrimental for NMR measurements (triethyl amine, remaining after HPLC) by centrifugation on Amicon filters in the excess of 10 mM cacodylate

buffer (pH 6.0) that was used during all subsequent experiments. The cleavable constructs were instead used directly as provided by the supplier to avoid any premature cleavage during a similar pretreatment.

### Kinetic assays
Kinetic assays were conducted to measure the catalytic activities for (1) the truncated 8–17_short construct and (2) a reference full-length bimolecular 8–17 DNAzyme construct from the crystallographic study (Supplementary Figures 2 and 3) in the presence of $Zn^{2+}$, $Mg^{2+}$ and $Pb^{2+}$. Both constructs were 3'-FAM labeled and contained a single RNA residue at the cleavage site. The dried samples purchased from Metabion were dissolved in the reaction buffer (10 mM sodium cacodylate buffer, pH 6.0) to 40 μM concentrations and used without further purification to minimize premature cleavage. In case of the bimolecular variant the FAM-labeled substrate and DNAzyme strands were used in 1:1 concentration ratio. Kinetic assays were performed at 25 °C. The reactions were initiated by adding the appropriate amounts of metal ions from stock solutions. Aliquots were collected at appropriate time points and quenched using a stop buffer containing 0.5 M EDTA and 8 M Urea. Samples were resolved by electrophoresis on a 15% urea gel and scanned on a Amersham Typhoon phosphorimager. The extent of reaction at each time point was quantified using Image-QuantTL software, version 10.2. The observed rate constants $k_{obs}$ were obtained by fitting the observed fraction cleaved ($y$) against time ($t$) to the equation: $y = y_0 + A\left(1 - e^{-k_{obs}t}\right)$. The reaction was repeated three times in each of the tested conditions presented in Supplementary Fig. 2 and the error bars represent the standard deviations of the fraction cut at each time point between these repetitions. The dependencies of the measured $k_{obs}$ values on metal ion concentrations (Supplementary Fig. 3) were fitted using a 1:1 interaction model to estimate the apparent $K_d$ values for metal ion binding. For this figure the $k_{obs}$ values at each metal concentration were derived in the same manner, however only for the 8–17_short interaction with $Mg^{2+}$ the experiments were done in triplicate, for the other 5 series each condition was tested only once (due to excellent reproducibility observed between the three replicates in the 8–17_short-$Mg^{2+}$ series that was performed as the first).

### CD spectroscopy
The circular dichroism spectra (CD) were recorded using a JASCO J815 spectropolarimeter equipped with a Peltier temperature controller and operating on JASCO Spectra Manager software version 2.09.11. Cuvettes with a path length of 0.1 cm were used. Spectra were collected in the range between 215 and 320 nm, as a sum of seven

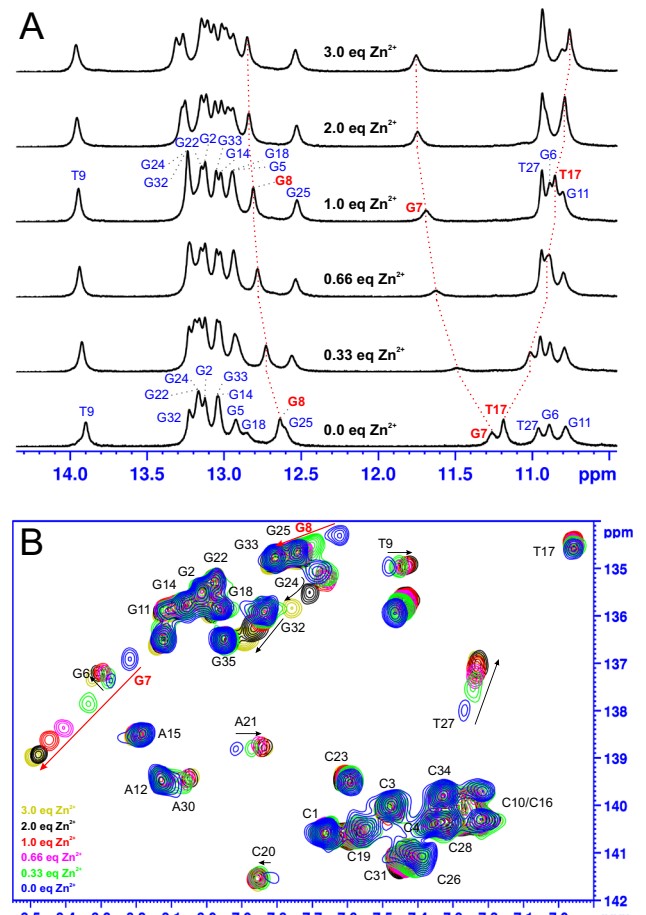

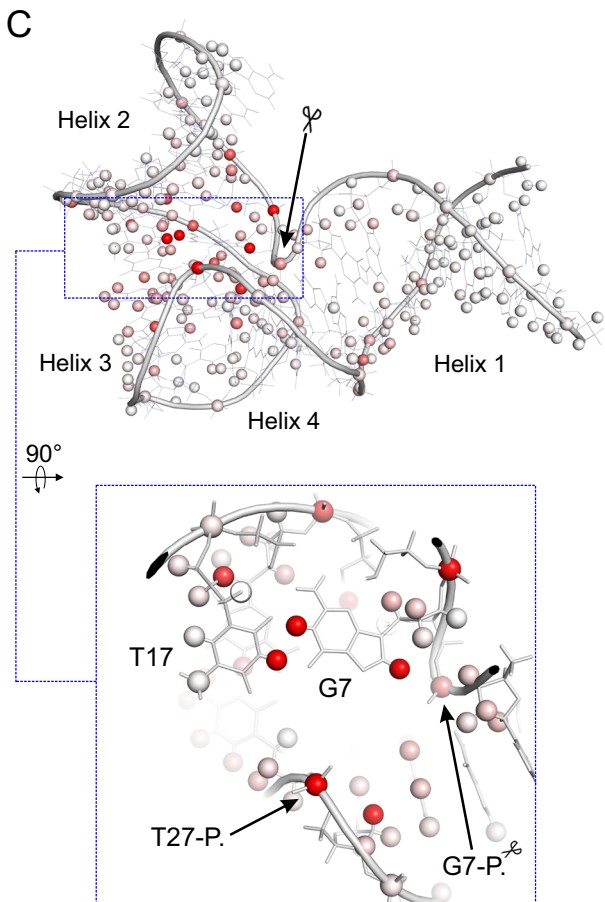

**Fig. 9 | The study of Zn²⁺ binding to Na⁺-structured 8-17_short through chemical shift perturbations (CSPs). A** spectral changes in the imino region of 1D ¹H-NMR spectrum upon Zn²⁺ titration, (**B**) spectral changes in the aromatic region

of ¹H-¹³C HSQC, (**C**) measured CSPs visualized on the 3D structure, color coded from 0.00 [ppm] (white) to ≥0.25 [ppm] (red). Source data are provided as Supplementary Data files.

repetitions at 20 °C and the buffer baseline was subtracted from each spectrum. The sample concentrations were adjusted to always give rise to absorbance values close to 1. This resulted in 40 μM samples for 8−17_short (similar to what was used in ref. 11) and around half that concentration for the full-length constructs. CD spectra were expressed in the units of molar ellipticity Δε [cm² mmol⁻¹], without normalization by the number of residues in the molecule.

### NMR spectroscopy

The NMR spectra were primarily collected using a 700 MHz Bruker Avance III spectrometer equipped with a QCI-P CryoProbe using the Topspin 3.6.5 software. Only the ³¹P-observed HP-COSY spectra were recorded on a 500 MHz Bruker Avance III spectrometer using a room-temperature BBO probe using Topspin 3.6.2, instead. The resonance assignment of non-exchangeable protons and phosphorus atoms was achieved using standard procedures[34,35], through the analysis of NOESY (with water suppression through excitation sculpting[36] and 80 or 150 ms mixing times), TOCSY[37], HC-HSQC[38] and HP-COSY[39] spectra recorded in 100% D₂O at 20 °C and 35 °C when possible. The exchangeable protons were assigned using NOESY spectra (with WATERGATE water suppression[40] and 80 or 250 ms mixing times) measured in 90% H₂O/10% D₂O at 5, 12.5, and 20 °C. The full range of sample conditions used are provided in Supplementary Data 1 and 2. All the spectra were analyzed in NMRFAM-Sparky[41] version 1.470. The resonance assignments were gathered in Supplementary Data 1 and 2, while snippets of assigned spectra can be found in Supplementary Figs. 7 and 8. All the reported chemical shift

perturbations (CSPs) were calculated by simple subtraction of ¹H chemical shifts measured at different points of the titrations.

### Restraint generation and NMR structure calculations

The structure determination of 8−17_short was driven by the classical NOE-derived distance restraints and dihedral angle restraints. The distance restraints between non-exchangeable protons were extracted from the peak volumes in NOESY spectra recorded in D₂O with and 150 ms mixing time. The NOESY cross-peaks were classified as strong (1.8–3.0 Å), medium (2.0–4.0 Å), weak (2.2–5.0 Å) and very weak (2.4–6.0 Å), using fixed distances (H5-H6 in cytidines and H2'-H2'' in deoxyriboses) as reference. Any peak that was not reliably integrable due to partial overlap was assigned to the broadest 1.8–7.0 Å category. The distance restraints between exchangeable protons were derived from the NOESY spectra recorded in H₂O (150 ms mixing) using a similar procedure, with the mean volume of the imino-amino contacts of guanosines used as reference. Regarding dihedral angle restraints, the backbone dihedral angles α and ζ were restrained to exclude the *trans* rotamer for all residues with ³¹P chemical shift within the standard range. Moreover, for residues for which the P(n)-H4'(n-1) cross peak was observable the β and γ dihedral angles were restrained to 180° and 60°, respectively (±60°). The χ dihedral angle was in turn restrained to the *anti* orientation for all residues based on the intensities of H1'-H6/H8 NOESY cross-peaks. For the residues with $J_{H3'-H4'} \approx 0$ Hz sugar puckers were restrained to the C2'-*endo* conformation. Hydrogen bond restraints for G·C base pairs were imposed based on the observation of strong NOE cross-peaks between specific cytidine

$NH_2$ and guanosine imino protons. For the C22-G28 base pair the assignment of the imino proton was confirmed using site-specific isotopic labeling (Supplementary Fig. 5a). Overall, a total of 365 NOE, 80 hydrogen bond, and 156 torsion angle (including 21 sugar pucker) restraints were obtained.

The structure calculations were performed using a two-step procedure, involving (1) initial folding from random extended conformations, in implicit solvent (generalized Born model) (2) refinement of the best pre-folded structures in explicit solvent. Both stages were executed in the SANDER module of the AMBER 18 molecular dynamics suite of programs[42]. All calculations used the parm99bsc0χ$_{OL15}$ force field[43] with SPC/E water model and Joung-Cheatham parameters for monovalent cations[44]. Long-range electrostatics were calculated using the particle mesh Ewald method with the nonbonded cutoff set to 8 Å. The covalent bonds were constrained using SHAKE and the integration time step was set to 2 fs. The Langevin thermostat with collision frequency 1.0 ps$^{-1}$ was used to control the temperature and Berendsen barostat for constant pressure simulation. In the first step a total of 200 structures were calculated using a simulated annealing protocol designed to enhance conformational sampling by significantly reducing the strength of the electrostatic and van der Waals components of the force field during the high temperature stage. Best among these structures were then solvated and refined using a much milder simulated annealing scheme in explicit solvent. In this final step 200 structures were calculated and then 20 were selected based on their reproduction of the experimental NMR data (Supplementary Table 1). All structure visualizations presented in this work were prepared in Pymol Molecular Graphics System, Version 1.6, Schrödinger, LLC.

### Reporting summary

Further information on research design is available in the Nature Portfolio Reporting Summary linked to this article.

## Data availability

Atomic coordinates and the list of experimental restraints for the reported NMR structure have been deposited with the Protein Data bank under accession code 8OR8 while the chemical shifts have been deposited at the BMRB under the numbers 34805 https://doi.org/10.13018/BMR34805 and 52355 https://doi.org/10.13018/BMR52355. The CD spectra and cleavage assays (gel scans) generated in this study are provided in the Source Data file. The raw NMR data used in this study are available in the Zenodo database under https://doi.org/10.5281/zenodo.11047570. This study also makes use of the previously published crystallographic structure: 5XM8 for structural comparison. Source data are provided with this paper.

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

## Acknowledgements

This work was supported by the National Science Center (Poland) under grant agreement: [UMO-2018/31/D/ST4/01467] to W.A. The calculations were performed at Poznań Supercomputing and Networking Center. The authors would like to thank Dr Karolina Zielińska for help in preparing the manuscript figures.

## Author contributions

W.A. designed the research. W.A. and Z.G. supervised the research. J.W., A.P., E.P., and K.P. collected the data, W.A., J.W., and A.P. analyzed the data, W.A. wrote the manuscript with comments and contributions from all other authors.

## Competing interests

The authors declare no competing interests.
