## [Peer Review File · Nature Communications]

The 8-17 DNAzyme can operate in a single active structure regardless of metal ion cofactorREVIEWER COMMENTS

Reviewer #1 (Remarks to the Author):

The authors use NMR spectroscopy to study the metal-dependent structure of the RNA-cleaving 8-17 DNAzyme, which is one of the two most well-studied such DNAzymes (the other is 10-23). This is done in the context of considerable biochemical studies already done on 8-17, along with a recent (2017, ref. 14) X-ray crystallography study that was the first structural report of any RNA-cleaving DNAzyme. These prior studies have led to the currently accepted view of 8-17, well summarized in Figure 1, that Pb²⁺ binds to a more open, less compact folded form of 8-17 than do most other divalent metal ions such as Zn²⁺ and Mg²⁺.

Here the authors study the Zn²⁺-bound 8-17 (also Na⁺-bound version), finding a structure surprisingly similar to the previous Pb²⁺-bound version. With these data, they propose a new model (Figure 9) in which 8-17 folds into only a single kind of catalytically active structure regardless of which metal ion is present, although the fractional population of this structure depends on the metal. A key feature of this new model is that Zn²⁺ and other divalent metal ions like Mg²⁺ can actively induce the catalytically active folded structure, whereas Pb²⁺ can bind to that structure but not induce its formation. Another key feature of the authors' model is that a high-affinity (micromolar) Zn²⁺ binding site that is required for folding, but is remote from the catalytic site, must be distinguished from a low-affinity (millimolar) Zn²⁺ binding site required for catalysis.

The manuscript is well-written, and in my view the authors do a proper job of noting some caveats to their conclusions. For instance, in the Discussion on page 12, paragraph beginning on line 317, the authors note that the low-affinity catalytic Zn²⁺ site does not appear to be populated in their experiments despite the millimolar Zn²⁺ concentration, due either to competing Zn²⁺ sites or the unavoidable use of a noncleavable construct lacking the 2'-OH at the scissile phosphate.

While I suspect that some readers will interpret the authors' data in other ways and thus not fully agree with the authors' model, I support publication of this interesting work in Nature Communications. In my view the authors have provided a strong set of data and an intriguing structural model to explain their data. This manuscript will certainly be the important starting point for a variety of experimental and computational studies of the 8-17 DNAzyme.

(This reviewer is well qualified as a nucleic acids chemist and biochemist but is not specifically an NMR spectroscopist or MD simulations expert. I have done my best to assess the technical aspects of these data, finding no particular faults, but perhaps an NMR or MD expert will find something to comment.)

Individual comments:

1. Regarding the lack of 2'-OH at the scissile phosphate, a natural question is to ask what would happen if 2'-OH were replaced not with 2'-H but with 2'-OMe (or possibly 2'-F), which should be experimentally

tractable. I do not consider such an experiment mandatory for publication, but the results using a 2'-OMe construct would probably be interesting and useful.

2. Regarding the inability to test Pb²⁺ binding to the NaCl-stabilized folded state due to PbCl₂ precipitation (page 12, line 343), would using any counterions other than chloride enable the intended testing?

3. Perhaps there is no good way to do this, but it would be great if the model in Figure 9 can incorporate the difference between the high-affinity (micromolar) Zn²⁺ binding site required for folding but remote from the catalytic site, versus the low-affinity (millimolar) Zn²⁺ site required for catalysis. Currently the model just depicts one kind of Zn²⁺ binding, and which kind is ambiguous. Minimally, the model (or the figure caption) should clarify this distinction. I think the authors' intention is that the illustrated Zn²⁺ ion corresponds to the higher-affinity structural Zn²⁺, not the lower-affinity catalytic Zn²⁺, where the authors make clear at several points in the manuscript that their current data do not locate the latter ion.

4. Page 2, line 46: "Both techniques concluded that Pb²⁺ binds directly to the apo- form of the enzyme and the catalytic reaction proceeds from this state." In the context of Figure 1d, this text description can be clarified. E.g., "Both techniques concluded that Pb²⁺ binds directly to the apo form of the enzyme without substantially changing its open structure, and the catalytic reaction proceeds from this state."

5. The Abstract does not state the experimental method (NMR spectroscopy) used for the DNAzyme structure determination.

6. Is ref. 3 ever cited in the manuscript text?

7. Please label the construct in Figure 1c as 8-17_{short}, as first noted in text on page 4, line 131.

8. The long paragraph crossing from page 4 to 5 would benefit from adding a paragraph break, such as on page 4, line 136, when switching to talk about the Zn²⁺-induced folding by NMR and CD spectroscopies.

9. Page 12, line 330, fix typo "yet with one is it" -> "yet which one is it".

10. Page 14, line 399, fix typo "7-deaze-dG" -> "7-deaza-dG" (is correct in Figure S6 caption).

11. Throughout the manuscript, I don't think the hyphen is needed when writing "apo- form". Just "apo form" is sufficient and correct.

12. This will presumably be fixed in most places by the copyeditor, but please check for unnecessary and distracting commas throughout the manuscript. Example: page 14, line 400, "as well as, their cleavable".

13. SI: please number the pages.

14. SI page 2, line 4, "significantly hurdled" -> "significantly burdened".
15. SI page 3, upper section, the text suddenly ends, "yielded a Kd value of 31 uM with".
16. SI page 3, last paragraph, "which we surprisingly did not observe" - it is not clear that "surprisingly" is appropriate here. The CD data were always not so compelling for the Z-DNA conclusion. As the authors note, it was based merely on a weak negative band at >290 nm, and other explanations are possible.
17. SI Figures S2 and S6, please consider making font sizes larger throughout.

Reviewer #2 (Remarks to the Author):

The manuscript by Wieruszewska et al. reports the first NMR structure of the 8-17 DNAzyme in solution. The results presented are of very high scientific quality and the insights gained are very exciting. The NMR structure was determined in the presence of Zn²⁺ and overall is very similar to the previously published crystal structure of an 8-17 DNAzyme variant in the Pb²⁺-bound conformation. This is an important result since it was suspected that different divalent metal ions can give rise to different conformations. The presented manuscript provides compelling data that can partially resolve conflicting previous data.

The authors use their findings to propose a new general mechanism for the involvement of divalent metal ions in the 8-17 DNAzyme based on the stabilization of different conformations in dynamic equilibrium. The new model is consistent with the new structural findings and with most previous results.

In addition to some technical aspects, my central concern with the presented results and the proposed model is the validity and general applicability of the proposed model, which is a key element of the manuscript. Therefore, the following aspects need to be addressed in a revised version either by providing additional data or by reformulating the respective interpretations and shifting the focus away from the proposed model. Because the outcome of possible new data (either confirming or modifying the proposed model) is unclear, it is difficult to assess at this time whether publication of a revised version in Nat. Comms. or a more specialised journal would be most suited (also see below).

Major points:

1. in general, the determined structure is very similar to the already known structure of the 8-17 DNAzyme. While this is an important finding, it naturally impacts the novelty aspects of the determined structure. Consequently, the authors focus on the differences between Pb²⁺ and Zn²⁺ binding, which is indeed probably the most important aspect of the manuscript. In general, the data convincingly show that Pb²⁺ and Zn²⁺ binding is realized by nearly identical structural frameworks. Conceptually, larger portions of the manuscript subsequently rely on literature data suggesting different structural effects of Pb²⁺ and Zn²⁺. In general, many of the relevant previous studies were performed under different

experimental conditions (including pH, buffers, temperatures, DNAzyme constructs, concentrations of DNAzymes and cofactors, and construction of structural models from low-resolution data). Thus, in many respects it is difficult to perform a reliable comparative analysis, which is, however, a central element of the present manuscript.

To take just one example: The mentioned aspect that previous studies have shown that the addition of Na^+ increases Pb^{2+} -mediated activity but not Mg^{2+} -mediated activity (lines 364-367) would indeed be a strong argument in favor of the proposed model. However, the corresponding study (ref. 19) was performed with a 'pKa-altered' DNAzyme variant and different pH values for the Mg^{2+} and Pb^{2+} experiments and shows not only an increase in activity but also a (slight) decrease in activity with further increasing NaCl concentrations. Moreover, the Mg^{2+} concentration used (20 mM) is much higher than the Pb^{2+} concentration (100 μM), which greatly reduces the molar access of Na^+ , which may well influence the observed effects. Thus, I find it difficult to use this study as support for the proposed model. Similarly, a number of conclusions are taken from other previous studies, which may not be fully compatible (or may have even been misinterpreted in the original publication).

Therefore, in order to strengthen (or modify) the proposed model, it is essential to conduct an independent set of experiments on the constructs under study and under comparable experimental conditions. In this regard, I would recommend including (at least) the following set of experiments :

- a. Activity assays involving NaCl titration at fixed Pb^{2+} as well as Zn^{2+} concentrations to evaluate the hypothesis that Zn^{2+} can induce the activated conformation but Pb^{2+} cannot.
- b. An NMR study of at least one additional divalent metal ion (e.g., Mg^{2+}) to evaluate the general applicability of the proposed model.
- c. An NMR study of Pb^{2+} . As mentioned by the authors, the solubility of Pb^{2+} will complicate this setup, but acquisition of basic NMR data should be possible at a DNAzyme concentration in the range of 10 μM , which should allow Pb^{2+} interaction studies. This would greatly strengthen the proposed differences in Pb^{2+} and Zn^{2+} effects.

2. The experimental design uses several "simplifications" of the system that might limit the general applicability of the obtained findings. While the authors convincingly show that the reduced arm length has no negative effect on activity, other factors clearly do. In particular, the use of low pH (pH 6, chosen to increase NMR accessibility of exchangeable protons) reduces activity by at least an order of magnitude (compared to neutral pH). In addition, the use of an all-DNA substrate results in a change in arm configuration from a presumed A-form (native RNA substrate in complex with DNAzyme) to a B-form helix, which is also known to greatly reduce activity (when using the cleavable DNA/RNA substrate). These factors may affect the general validity of the findings obtained with respect to the "standard" applications of the 8-17 DNAzyme (all RNA substrates at neutral pH). Therefore, this must be adequately discussed in the manuscript as a potential shortcoming of the approach used, or additional NMR and activity data recorded under the respective "standard" conditions must be included to confirm the validity of the respective findings under these conditions.

3. Please provide information about the used NaCl concentration for data in Fig. 3. In case 200 mM NaCl was used (as to be expected from the subsequent text), please check why the NMR spectrum in Fig. 3B (0 mM Zn^{2+} , 200 mM NaCl?) has noticeable differences to the spectrum in Fig. 6B (200 mM NaCl, 0 mM Zn^{2+}) that largely exceed the expected experimental errors.

4. While the discussed model does account for previous ensemble techniques, it does not adequately discuss the single molecule FRET data, which, to my understanding, would not be well compatible with this model. Please include a brief discussion of this aspect.

Minor aspect:

5. Please show the NMR ensemble already in Fig. 4.

6. It is very difficult to see the discussed features in Fig. S4. It seems that there are significant differences between the only Na⁺ and only Zn²⁺ states that are not discussed. In addition, the "similar" peak shifts are not clearly visible. It would be very helpful to improve the corresponding figures, e.g., to include pairwise overlays and to highlight (only) selected peaks. Furthermore, it would probably be easier to make (additional) comparisons between the Zn²⁺-only conditions and the other two conditions by using the 1D plots already used for the other two conditions (Fig. 3 and Fig. 6).

7. It would be helpful to discuss more explicitly the similarities and potential differences between the Pb²⁺-bound structure and the Zn²⁺-bound structure, including whether the GG kink at the cleavage site and the proposed "L-platform" are also present in the NMR structure. In addition, a specific schematic comparison (e.g., moving Fig. 2B to a later figure and adding a corresponding model of the Zn²⁺-bonded state) would help to highlight similarities and potential differences between the two structures.

8. The caption of Fig. 6 should probably read "Na⁺ induced folding...".

9. Line 330, "with"  "which"

Overall, I would like to emphasise that this is an excellent work that provides important new results that are very convincing in their own right. My main concern is the interpretation of the data in the context of a new generalized model that relies largely on a comparative analysis of previous results from different laboratories recorded under different conditions. The addition of further data under fully comparable conditions (point 1) could significantly strengthen the proposed model and allow publication in Nat. Comms. Nevertheless, the data already included in the current version could also be used for publication, but this would require a shift in focus from the proposed model and general applicability to more specific results and consequently would be more suitable for a more specialised journal.

Reviewer #3 (Remarks to the Author):

This manuscript reports a solution NMR structure of a short-armed single molecule construct of the 8-17 DNAzyme resolved in presence of Zn²⁺ ion.

Previously it was shown through FRET studies that the 8-17 DNAzyme goes through further folding in

presence of metal ions such as Zn^{2+} but not with Pb^{2+} . It is of great significance to understand the structural differences and requirements for the various metal ions to get a better understanding of the mechanism of this DNAzyme, and its dependence on the metal cofactor.

This paper, based on the title and abstract, claims to have established that there is “single active structure regardless of the metal ion”, however falls short on the delivery.

The title is misleading, the paper does not have enough data to claim there is “single active structure regardless of the metal ion”. The study has only resolved a structure in the presence of one type of metal ion, Zn^{2+} , and this was done with a different construct than the available crystal structure, so there aren't even two comparable points to make a line.

The abstract claims “high resolution solution structure of 8-17 DNAzyme bound to a Zn^{2+} ion”. This is misleading. Many readers will, as this reviewer has, interpret this as the catalytic Zn ion was resolved bound at the active site. It is not even immediately obvious from the abstract that this is an NMR structure, aside from the “solution” keyword, which comes right after the word “resolution” and is easily skipped by the eye upon first read. The deposited structures do not contain any metal ions, and later in the text this is revealed that a “structural” Zn ion binding site was discovered.

Similarly “Our results constitute the first high-resolution structural information on the 8-17 DNAzyme in presence of non- Pb^{2+} cofactors, including the biologically relevant Mg^{2+} ion”, is misleading. The paper does not report any results obtained with Mg^{2+} ion, but only extrapolates the Zn^{2+} results.

It is clear that this new structure has a catalytic core arrangement that is very similar to the available crystal structure. It is not clear what the effect of the new construct is in obtaining/sustaining this, as no control experiments of rate, CD or NMR are reported for the WT two-strand system for comparison. It is clear that the short-armed construct has a better rate performance than the full-length (still capped) construct, so the authors are already working with a “more active” version of their capped system.

To make the claim “single active structure” there would need to be additional structures resolved in the presence of various metal ions especially Pb^{2+} (as a control), and Mg^{2+} that yield the same structure. In its current form, the paper makes a lot of big claims without the experiments to support them. With the experiments that it does report the paper is more appropriate for a more specialized journal.

Beyond the content, there are various aspects that need improving for publication anywhere. Some of them are listed below in order of their appearance in the manuscript.

1. Overall issues with clarity regarding which statements are facts, supported by the experimental data current or previous, and which statements are assumptions or interpretations (and in places extrapolations).
2. Lines 77-78. The statement “initial step” is not supported. Currently there is no evidence regarding the order of the chemical steps, nucleophile activation doesn't necessarily need to be the “initial” step.

3. Lines 95-96: “validated for unaltered catalytic activity”. How is it unaltered? Where’s this evidence? There is no comparison to WT two-strand struct. Clearly the rate is different between their short and full-length structures.

4. Lines 152-153: “we will use the numbering specific to this construct”. This made it really hard to follow the arguments in the paper as only new residue numbers were given without any context. Authors should include canonical numbers in parentheses and state the role of the residue when necessary (i.e. G7 (G1) upstream from scissile phosphate)

5. Lines 168-170: It’s not clear what is meant by this statement. These are 3 base-pairs, 2 of which are non-canonical that form the L-platform motif commonly observed in the active sites of RNA-cleaving nucleic acid enzymes. It is not clear how other than the crystal structures these would have been observed, and what the emphasis of the statement here is. Consider rephrasing. If anything should be emphasized here it’s that these base-pairs form the L-platform motif [Gaines RNA 2020. 26: 111-125].

6. Lines 198-201. In its current form it is not convincing that the Na⁺ induces the same fold as Zn²⁺. The CD data are significantly different. There’s no discussion of what kind of similarity in the NMR results would yield in what kind of structural similarity.

7. Lines 214-217. A good example of the concern in #1. Is this an assumption or a statement based on the data seen? (either way clarify, describe). It does not make chemical sense that a displacement of a Na⁺ ion (+1 charge with a large ionic radius), with Zn²⁺ ion (+2 charge and small radius), does not induce structural changes. Is it meant that it doesn’t affect the overall structure?

8. Lines 305-308. “The fact that for 8-17_short enzymatic rate k_{obs} Zn²⁺ does not saturate with the DNAzyme folding, but continues to increase with concentration well Zn²⁺ into the millimolar range (Figure S7), strongly suggests that a different ion – binding with a much lower affinity – is actually the one required for catalysis”. Isn’t this a different behavior than the WT, and might possibly suggest a different mechanism for the current construct?

9. Lines 343-345 “Unfortunately, we were not able to test Pb²⁺ binding to the NaCl stabilized folded state in solution, as in these conditions the lead ion would precipitate as PbCl₂, a very weakly soluble salt (K_{so}= 1.7×10⁻⁵ at 20 °C).” This statement is confusing. What are “these” conditions? Would the system not carry out catalysis if Pb²⁺ were be added? It also is never explicitly stated what Na-stabilized system is, is it 400 mM NaCl?

10. Lines 353-354. "Pb²⁺ only binds to whatever amount of this state is initially present in the conformational ensemble..." This is inconsistent with all the previous functional data. This would result in very a very low %cleavage with Pb compared to Zn, which is not the case [Woohyun Biochemistry 2021. 60: 1909].

11. Lines 364-366. Providing a compelling structural explanation here is not sufficient. There’s already a chemical explanation for this behavior. The observation in Ref 19 regarding the effect of Na⁺ with Pb²⁺ is not only a structural one, but also a chemical one. Ref 19 found that both the acid-base pK_a’s and the

reaction rates were affected by Na⁺ concentration. It was a test of the computational findings of Ref 20, which showed that Na⁺ ions facilitated the active in-line state by bridging the O2' and the pro-Rp. Without mentioning the background this it seems like their hypothesis is the only explanation currently available for this observation.

12. Lines 365 – 366. “the catalytic reaction of 8-17 DNAzyme 365 in the presence of Pb²⁺ (which cannot fold the active state on its own”. This is a major misinterpretation. The fact that further folding wasn't seen in the concentrations tested does not mean that Pb cannot induce further folding. The following segment from Ref 3 clearly states so. “Since the concentrations of Pb²⁺ used in this investigation (0–100 μM) are much lower than those of Zn²⁺ (0–500 μM) and Mg²⁺ (0–20 mM) due to the Pb²⁺ precipitation problem at higher concentrations, we cannot rule out the possibility that the DNAzyme will fold at much higher Pb²⁺ concentrations (if one can solve the precipitation problem).”

13. Lines: 377-378. “these conclusions can be extended to catalysis in the presence of Zn²⁺ and other divalent 377 metal ions, notably Mg²⁺”. This is not accurate. Mg²⁺ and Pb²⁺ have different mutational signatures for G6 (G22), so clearly something regarding the metal ion binding is different.

14. Molecular dynamics approach is faulty in the current way it is stated. Lines 471 – 475 “The first five conformers from the NMR bundle were solvated using octahedral box of SPC/E water with a minimum distance between box walls and solute of 10 Å and additional Na⁺ ions were added to neutralize the system. Each system was prepared twice, with one copy containing a Zn²⁺ ion manually introduced before solvation, yielding a total of 10 independent simulation setups. No additional salt was added.”

a. The NMR structures do not have the nucleophile and are therefore inactive. The MD should have been done on the reactive system. It's one thing to collect NMR data from an inactive system (assumably for signal clarity, although that is also not mentioned) but what is the reasoning behind doing MD on the inactive system?

b. The only ions in the system were the charge neutralizing (counter) ions, which for this small system would be ~36. This means the Na concentration for MD was 0 (zero). One cannot interpret “binding sites” in a simulation with 0 bulk concentration of ions. Na⁺ parameters are the best available ion parameters for molecular simulations and Na⁺ ions can easily exchange and bind to different nucleotides within simulations especially in the microsecond scale, therefore there is no excuse for the lack of a significant concentration bulk Na⁺ ions in these simulations. At least 0.140 M NaCl should have been included. Especially for a system that requires a very high concentration of Na⁺ ions in solution. The fact that two sites were occupied >90% suggests those ions were stuck there. This could very well be due to lack of free floating ions to exchange with.

c. Where was the Zn manually placed? Did it move? Was the total charge of the Zn systems +2? Or were 2 Na⁺ removed to neutralize the charge? Or 2 Cl⁻ were added? All these info need to be provided.

Figures:

- All figure legends lack necessary descriptions.

- Font sizes are inconsistent. Some inner panels are not legible. Some descriptive figures have very large

and very small fonts, making it hard to follow (i.e. having to zoom in and out of the figure to read different sections).

- Figure 6 legend says Zn, but the figure is Na.

General information for Reviewers - Overview of the additional experimental results

As many additional experiments and analyses were performed following the suggestions of the Reviewers, we find it useful to provide a brief overview of the newly obtained results, before addressing the specific points risen by the Reviewers.

The newly added experimental data include:

- in depth structural analysis of the short variant of the 8-17 DNAzyme (8-17_short) in presence of Mg^{2+} using 1D and 2D NMR, as well as CD spectroscopies
- same set of experiments and structural analysis performed for 8-17_short in presence of Pb^{2+}
- Zn^{2+} , Na^+ , Mg^{2+} and Pb^{2+} titrations of the full-length, bimolecular variant of the DNAzyme used previously for X-ray crystallography monitored by 1D NMR and CD spectroscopies
- several series of activity assays for both the short and full-length variants in presence of various concentrations of Zn^{2+} , Mg^{2+} and Pb^{2+} , as well as some combinations of these ions to obtain the k_{obs} vs metal concentration profiles
- 1D NMR and CD-monitored titrations of 8-17_short with Zn^{2+} at pH 7

As described in detail in the article text and SI most of the new experiments produced results fitting into the picture drawn by our original results and previous reports. To name a few among these results: Mg^{2+} induced a folding transition to a global fold similar to the one we observed for Zn^{2+} , K_d values for interactions with metal ions were comparable with the previous literature, spectral changes induced by Zn^{2+} at pH 7 mirrored those observed at pH 6. However, the spectroscopic investigations performed in the presence of Pb^{2+} produced a largely unexpected outcome. We have observed global folding of the DNAzyme even by this ion alone at both NMR and CD concentrations and for both the short and full-length 8-17 variants. In depth structural analysis of the NMR data for the short construct revealed that we were once again dealing with the same global fold as we originally observed with Zn^{2+} . Experiments on the full-length construct also shown all four tested ions - Zn^{2+} , Na^+ , Mg^{2+} and Pb^{2+} - to induce similar CD and 1D NMR spectral changes, yet in this case we were not able to perform a deeper 2D NMR study due to the system being too large. Overall, these results – while unexpected – reinforce our previous suggestion that the 8-17 DNAzyme may only have a single active conformation, regardless of the metal ion cofactor, yet they also rise even more questions regarding why similar behavior was not reported before, which we try to discuss in length in the updated paper's SI.

The observation of Pb^{2+} -induced folding required a revision to be included in our newly proposed model for metal ion interactions with 8-17 DNAzyme. We originally proposed that Pb^{2+} cannot bind to the “structural” binding site of the DNAzyme, yet has a high affinity for the “catalytic” site. Now we have experimentally estimated the Pb^{2+} K_d values for the “structural” and “catalytic” sites through CD spectroscopy and activity assays, respectively and found them to be of a similar order of magnitude. Thus, in our updated model for metal ion interactions with 8-17 DNAzyme we allow Pb^{2+} to also interact with the “structural” site and fold the DNAzyme, yet with the “catalytic” site being filled alongside it, resulting in significant catalytic activity being observable even at Pb^{2+} concentration at which the DNAzyme remains largely unfolded. Such behavior still makes Pb^{2+} distinct among other metal ion cofactors, that require orders of magnitude higher concentrations to effectively induce catalytic activity, as compared to what is required for global folding.

The addition of the new data also urged us to switch to a “Results and discussion” format so that each new unexpected result can immediately be contextualized by discussing the relevant literature. Of the previous three sections of the discussion:

- the first, discussing the geometry and function of the high affinity Zn^{2+} binding site, was integrated with the corresponding section originally in the Results section
- the second one, proposing the new model, was moved just after the section describing the Pb^{2+} induced folding and partially rewritten
- the last, discussing what our results add to the understanding of 8-17 catalysis, was renamed to Conclusions

REVIEWER COMMENTS

Reviewer #1 (Remarks to the Author):

The authors use NMR spectroscopy to study the metal-dependent structure of the RNA-cleaving 8-17 DNAzyme, which is one of the two most well-studied such DNAzymes (the other is 10-23). This is done in the context of considerable biochemical studies already done on 8-17, along with a recent (2017, ref. 14) X-ray crystallography study that was the first structural report of any RNA-cleaving DNAzyme. These prior studies have led to the currently accepted view of 8-17, well summarized in Figure 1, that Pb²⁺ binds to a more open, less compact folded form of 8-17 than do most other divalent metal ions such as Zn²⁺ and Mg²⁺.

Here the authors study the Zn²⁺-bound 8-17 (also Na⁺-bound version), finding a structure surprisingly similar to the previous Pb²⁺-bound version. With these data, they propose a new model (Figure 9) in which 8-17 folds into only a single kind of catalytically active structure regardless of which metal ion is present, although the fractional population of this structure depends on the metal. A key feature of this new model is that Zn²⁺ and other divalent metal ions like Mg²⁺ can actively induce the catalytically active folded structure, whereas Pb²⁺ can bind to that structure but not induce its formation. Another key feature of the authors' model is that a high-affinity (micromolar) Zn²⁺ binding site that is required for folding, but is remote from the catalytic site, must be distinguished from a low-affinity (millimolar) Zn²⁺ binding site required for catalysis.

The manuscript is well-written, and in my view the authors do a proper job of noting some caveats to their conclusions. For instance, in the Discussion on page 12, paragraph beginning on line 317, the authors note that the low-affinity catalytic Zn²⁺ site does not appear to be populated in their experiments despite the millimolar Zn²⁺ concentration, due either to competing Zn²⁺ sites or the unavoidable use of a noncleavable construct lacking the 2'-OH at the scissile phosphate.

While I suspect that some readers will interpret the authors' data in other ways and thus not fully agree with the authors' model, I support publication of this interesting work in Nature Communications. In my view the authors have provided a strong set of data and an intriguing structural model to explain their data. This manuscript will certainly be the important starting point for a variety of experimental and computational studies of the 8-17 DNAzyme.

(This reviewer is well qualified as a nucleic acids chemist and biochemist but is not specifically an NMR spectroscopist or MD simulations expert. I have done my best to assess the technical aspects of these data, finding no particular faults, but perhaps an NMR or MD expert will find something to comment.)

Individual comments:

1. Regarding the lack of 2'-OH at the scissile phosphate, a natural question is to ask what would happen if 2'-OH were replaced not with 2'-H but with 2'-OMe (or possibly 2'-F), which should be experimentally tractable. I do not consider such an experiment mandatory for publication, but the results using a 2'-OMe construct would probably be interesting and useful.

We agree that the presence of a 2'-OMe or 2'-F modified nucleotide at the cleavage site should in principle provide a more accurate model to study the local geometry around this site, by conserving the C3'- conformation likely adapted by the RNA nucleotide in an active construct. However, the previous crystallographic study reported the structures of both the 2'-H and 2'-OMe versions of their 8-17 DNAzyme construct (PDB: 5XM9 and 5XMA, respectively) and found little structural differences between them. We now added this information to the "8-17 DNAzyme construct optimization for NMR studies" section of the SI.

2. Regarding the inability to test Pb²⁺ binding to the NaCl-stabilized folded state due to PbCl₂ precipitation (page 12, line 343), would using any counterions other than chloride enable the intended testing?

Yes, a similar experiment would probably be possible by using for example NaNO₃ as the salt inducing the folding. However, the presence of a different counter ion would slightly change most of the chemical shifts of the Na⁺-stabilized fold (starting point for titration with divalent ions) and thus these spectra would have to be reassigned first. In the end we managed to resolve the question of Pb²⁺ binding to the "compact" state of the enzyme through titrations without any additional salt being added and observing Pb²⁺-induced folding (as described in the "General information for Reviewers ..." above), which demonstrated the interaction even more directly.

3. Perhaps there is no good way to do this, but it would be great if the model in Figure 9 can incorporate the difference between the high-affinity (micromolar) Zn²⁺ binding site required for folding but remote from the catalytic site, versus the low-affinity (millimolar) Zn²⁺ site required for catalysis. Currently the model just depicts one kind of Zn²⁺ binding, and which kind is ambiguous. Minimally, the model (or the figure caption) should clarify this distinction. I think the authors' intention is that the illustrated Zn²⁺ ion corresponds to the higher-affinity structural Zn²⁺, not the lower-affinity catalytic Zn²⁺, where the authors make clear at several points in the manuscript that their current data do not locate the latter ion.

We really like this suggestion by the reviewer, as including the two binding site in the figure makes it better represent our new proposed model. A new version of the figure was prepared, featuring the two binding sites, as well as additional panels to visually differentiate the sequential filling of the two sites, as proposed for Zn²⁺ from a simultaneous one as proposed for Pb²⁺.

4. Page 2, line 46: "Both techniques concluded that Pb²⁺ binds directly to the apo- form of the enzyme and the catalytic reaction proceeds from this state." In the context of Figure 1d, this text description can be clarified. E.g., "Both techniques concluded that Pb²⁺ binds directly to the apo form of the enzyme without substantially changing its open structure, and the catalytic reaction proceeds from this state."

The sentence was changes as suggested by the Reviewer.

5. The Abstract does not state the experimental method (NMR spectroscopy) used for the DNAzyme structure determination.

This information was added to the abstract.

6. Is ref. 3 ever cited in the manuscript text?

It is cited at page 1, line 30.

7. Please label the construct in Figure 1c as 8-17_short, as first noted in text on page 4, line 131.

Labels were added to 8-17_short and the crystallographic construct.

8. The long paragraph crossing from page 4 to 5 would benefit from adding a paragraph break, such as on page 4, line 136, when switching to talk about the Zn²⁺-induced folding by NMR and CD spectroscopies.

The paragraph break was added.

9. Page 12, line 330, fix typo "yet with one is it" -> "yet which one is it".

The typo was fixed.

10. Page 14, line 399, fix typo "7-deaze-dG" -> "7-deaza-dG" (is correct in Figure S6 caption).

The typo was fixed.

11. Throughout the manuscript, I don't think the hyphen is needed when writing "apo- form". Just "apo form" is sufficient and correct.

The hyphen was removed.

12. This will presumably be fixed in most places by the copyeditor, but please check for unnecessary and distracting commas throughout the manuscript. Example: page 14, line 400, "as well as, their cleavable".

We checked the punctuation throughout the manuscript.

13. SI: please number the pages.

Page numbers were added in the SI.

14. SI page 2, line 4, "significantly hurdled" -> "significantly burdened".

The suggested change was introduced.

15. SI page 3, upper section, the text suddenly ends, "yielded a Kd value of 31 uM with".

The missing conclusion of the phrase was added.

16. SI page 3, last paragraph, "which we surprisingly did not observe" - it is not clear that "surprisingly" is appropriate here. The CD data were always not so compelling for the Z-DNA conclusion. As the authors note, it was based merely on a weak negative band at >290 nm, and other explanations are possible.

We decided to keep the expression "surprisingly", as the Z-DNA explanation from the previous CD was cited multiple times in the subject literature and thus could be considered by many readers as the accepted explanation until now.

17. SI Figures S2 and S6, please consider making font sizes larger throughout.

The fonts in Figure S2 were increased. Regarding figure S6 (now S14) we assume that the Reviewer is referring to the fonts in the legends in panels presenting the CD spectra. We agree that these fonts are indeed difficult to read without magnifying the image, but unfortunately, we are unable to make them much bigger as in that case the legend would cover relevant parts of the spectra.

Reviewer #2 (Remarks to the Author):

The manuscript by Wieruszewska et al. reports the first NMR structure of the 8-17 DNAzyme in solution. The results presented are of very high scientific quality and the insights gained are very exciting. The NMR structure was determined in the presence of Zn²⁺ and overall is very similar to the previously published crystal structure of an 8-17 DNAzyme variant in the Pb²⁺-bound conformation. This is an important result since it was suspected that different divalent metal ions can give rise to different conformations. The presented manuscript provides compelling data that can partially resolve conflicting previous data.

The authors use their findings to propose a new general mechanism for the involvement of divalent metal ions in the 8-17 DNAzyme based on the stabilization of different conformations in dynamic equilibrium. The new model is consistent with the new structural findings and with most previous results.

In addition to some technical aspects, my central concern with the presented results and the proposed model is the validity and general applicability of the proposed model, which is a key element of the manuscript. Therefore, the following aspects need to be addressed in a revised version either by providing additional data or by reformulating the respective interpretations and shifting the focus away from the proposed model. Because the outcome of possible new data (either confirming or modifying the proposed model) is unclear, it is difficult to assess at this time whether publication of a revised version in Nat. Comms. or a more specialised journal would be most suited (also see below).

Major points:

1. in general, the determined structure is very similar to the already known structure of the 8-17 DNAzyme. While this is an important finding, it naturally impacts the novelty aspects of the determined structure. Consequently, the authors focus on the differences between Pb²⁺ and Zn²⁺ binding, which is indeed probably the most important aspect of the manuscript. In general, the data convincingly show that Pb²⁺ and Zn²⁺ binding is realized by nearly identical structural frameworks. Conceptually, larger portions of the manuscript subsequently rely on literature data suggesting different structural effects of Pb²⁺ and Zn²⁺. In general, many of the relevant previous studies were performed under different experimental conditions (including pH, buffers, temperatures, DNAzyme constructs, concentrations of DNAzymes and cofactors, and construction of structural models from low-resolution data). Thus, in many respects it is difficult to perform a reliable comparative analysis, which is, however, a central element of the present manuscript.

To take just one example: The mentioned aspect that previous studies have shown that the addition of Na⁺ increases Pb²⁺-mediated activity but not Mg²⁺-mediated activity (lines 364-367) would indeed be a strong argument in favor of the proposed model. However, the corresponding study (ref. 19) was performed with a 'pKa-altered' DNAzyme variant and different pH values for the Mg²⁺ and Pb²⁺ experiments and shows not only an increase in activity but also a (slight) decrease in activity with further increasing NaCl concentrations. Moreover, the Mg²⁺ concentration used (20 mM) is much

higher than the Pb²⁺ concentration (100 μM), which greatly reduces the molar access of Na⁺, which may well influence the observed effects. Thus, I find it difficult to use this study as support for the proposed model. Similarly, a number of conclusions are taken from other previous studies, which may not be fully compatible (or may have even been misinterpreted in the original publication).

Therefore, in order to strengthen (or modify) the proposed model, it is essential to conduct an independent set of experiments on the constructs under study and under comparable experimental conditions. In this regard, I would recommend including (at least) the following set of experiments :

- a. Activity assays involving NaCl titration at fixed Pb²⁺ as well as Zn²⁺ concentrations to evaluate the hypothesis that Zn²⁺ can induce the activated conformation but Pb²⁺ cannot.
- b. An NMR study of at least one additional divalent metal ion (e.g., Mg²⁺) to evaluate the general applicability of the proposed model.
- c. An NMR study of Pb²⁺. As mentioned by the authors, the solubility of Pb²⁺ will complicate this setup, but acquisition of basic NMR data should be possible at a DNAzyme concentration in the range of 10 μM, which should allow Pb²⁺ interaction studies. This would greatly strengthen the proposed differences in Pb²⁺ and Zn²⁺ effects.

The set of experiments proposed by the Reviewer was performed and the results integrated into the main text (and described in short above). The Pb²⁺ study (point c) was performed at the same high concentrations of the DNAzyme as similar studies for other cofactors. Our original remark regarding PbCl₂ precipitation regarded the titration of Pb²⁺ into a DNAzyme already structured by NaCl. We originally did not expect to observe any significant spectral changes when titrating Pb²⁺ into unstructured DNAzyme samples following previous literature observations.

2. The experimental design uses several "simplifications" of the system that might limit the general applicability of the obtained findings. While the authors convincingly show that the reduced arm length has no negative effect on activity, other factors clearly do. In particular, the use of low pH (pH 6, chosen to increase NMR accessibility of exchangeable protons) reduces activity by at least an order of magnitude (compared to neutral pH). In addition, the use of an all-DNA substrate results in a change in arm configuration from a presumed A-form (native RNA substrate in complex with DNAzyme) to a B-form helix, which is also known to greatly reduce activity (when using the cleavable DNA/RNA substrate). These factors may affect the general validity of the findings obtained with respect to the "standard" applications of the 8-17 DNAzyme (all RNA substrates at neutral pH). Therefore, this must be adequately discussed in the manuscript as a potential shortcoming of the approach used, or additional NMR and activity data recorded under the respective "standard" conditions must be included to confirm the validity of the respective findings under these conditions.

To verify whether the lower pH value influenced the results we obtain we have performed NMR and CD monitored titrations of 8-17_{short} with Zn²⁺ at pH 7. The results are presented in Figure S4 and are practically identical as those obtained during our original titrations at pH 6 (Figure 3). We thus added to the manuscript a conclusion that varying pH between 6 and 7 has no influence on the structural aspects of the DNAzyme's behavior.

Regarding the possible impact of the arms adopting the B-form conformation in our system, to best of our knowledge such a feature does not necessarily reduce the activity compared to systems with arms in the A-form. The comparison of cleavage rates between chimeric (DNA + 1 RNA nucleotide at the cleavage site) and all-RNA substrates provided in the 2009 review article (Schlosser and Li, ref 8 in the manuscript) gives examples of several 8-17 variants for which the cleavage of the chimeric substrate was actually more effective. Moreover, most previous investigations into 8-17 DNAzyme structure used all-DNA constructs and thus the use of a chimeric system allows us for a more direct comparison with previous results.

3. Please provide information about the used NaCl concentration for data in Fig. 3. In case 200 mM NaCl was used (as to be expected from the subsequent text), please check why the NMR spectrum in Fig. 3B (0 mM Zn²⁺, 200 mM NaCl?) has noticeable differences to the spectrum in Fig. 6B (200 mM NaCl, 0 mM Zn²⁺) that largely exceed the expected experimental errors.

No added NaCl was present during experiments presented in Figure 3. Given the context in the article (just before introducing conditions containing both Zn²⁺ and Na⁺) this can indeed be misleading and thus we added the appropriate information to the figure caption.

4. While the discussed model does account for previous ensemble techniques, it does not adequately discuss the single

molecule FRET data, which, to my understanding, would not be well compatible with this model. Please include a brief discussion of this aspect.

The discussion of how, in our opinion, the smFRET data can be interpreted in the light of our new findings was added to the SI in the section: "Previous FRET and CD spectroscopy results in the light the results reported in this work"

Minor aspect:

5. Please show the NMR ensemble already in Fig. 4.

We find it difficult to include the NMR ensemble without affecting the clarity of this figure. Instead, we added the information that the ensemble can be seen in Figure 5 much earlier in the text, just after introducing Figure 4.

6. It is very difficult to see the discussed features in Fig. S4. It seems that there are significant differences between the only Na⁺ and only Zn²⁺ states that are not discussed. In addition, the "similar" peak shifts are not clearly visible. It would be very helpful to improve the corresponding figures, e.g., to include pairwise overlays and to highlight (only) selected peaks. Furthermore, it would probably be easier to make (additional) comparisons between the Zn²⁺-only conditions and the other two conditions by using the 1D plots already used for the other two conditions (Fig. 3 and Fig. 6).

We added two new sections to the SI – "NMR spectral patterns observed for the 8-17 short in the structure determination conditions - long-range NOE contacts and uncommon chemical shift values" and "NMR observations for samples containing only a single metal ion type (Na⁺, Mg²⁺, Zn²⁺ or Pb²⁺)" – that discuss the 2D NMR data in different conditions in much more depth. A new panel was also added to old Figure S4 – now Figure S8 – showing the same spectral region as panel a of the original figure, yet with sequential connectivity pathways ("NOESY-walks") drawn to guide the reader's eye.

7. It would be helpful to discuss more explicitly the similarities and potential differences between the Pb²⁺-bound structure and the Zn²⁺-bound structure, including whether the GG kink at the cleavage site and the proposed "L-platform" are also present in the NMR structure. In addition, a specific schematic comparison (e.g., moving Fig. 2B to a later figure and adding a corresponding model of the Zn²⁺-bonded state) would help to highlight similarities and potential differences between the two structures.

Upon the Reviewer's inquiry we have delved deeper into the possible local structural changes between the folds of the catalytic domains found in crystallographic and NMR structures. We found that the only moiety for with a position difference beyond the uncertainty of the NMR bundle can be observed s the phosphate group of T27(N12). We added this information to the main text. We now also explicitly state in the Conclusions that the "L-platform" motif found in the crystals is also retained in the NMR structure.

8. The caption of Fig. 6 should probably read "Na⁺ induced folding...".

The caption was corrected.

9. Lline 330, "with"  "which"

The typo was fixed.

Overall, I would like to emphasise that this is an excellent work that provides important new results that are very convincing in their own right. My main concern is the interpretation of the data in the context of a new generalized model that relies largely on a comparative analysis of previous results from different laboratories recoded under different conditions. The addition of further data under fully comparable conditions (point 1) could significantly strengthen the proposed model and allow publication in Nat. Comms. Nevertheless, the data already included in the current version could also be used for publication, but this would require a shift in focus from the proposed model and general applicability to more specific results and consequently would be more suitable for a more specialised journal.

Reviewer #3 (Remarks to the Author):

This manuscript reports a solution NMR structure of a short-armed single molecule construct of the 8-17 DNAzyme resolved in presence of Zn²⁺ ion.

Previously it was shown through FRET studies that the 8-17 DNAzyme goes through further folding in presence of metal ions such as Zn²⁺ but not with Pb²⁺. It is of great significance to understand the structural differences and requirements for the various metal ions to get a better understanding of the mechanism of this DNAzyme, and its dependence on the metal cofactor.

This paper, based on the title and abstract, claims to have established that there is “single active structure regardless of the metal ion”, however falls short on the delivery.

The title is misleading, the paper does not have enough data to claim there is “single active structure regardless of the metal ion”. The study has only resolved a structure in the presence of one type of metal ion, Zn²⁺, and this was done with a different construct than the available crystal structure, so there aren't even two comparable points to make a line.

The abstract claims “high resolution solution structure of 8-17 DNAzyme bound to a Zn²⁺ ion”. This is misleading. Many readers will, as this reviewer has, interpret this as the catalytic Zn ion was resolved bound at the active site. It is not even immediately obvious from the abstract that this is an NMR structure, aside from the “solution” keyword, which comes right after the word “resolution” and is easily skipped by the eye upon first read. The deposited structures do not contain any metal ions, and later in the text this is revealed that a “structural” Zn ion binding site was discovered.

Similarly “Our results constitute the first high-resolution structural information on the 8-17 DNAzyme in presence of non-Pb²⁺ cofactors, including the biologically relevant Mg²⁺ ion”, is misleading. The paper does not report any results obtained with Mg²⁺ ion, but only extrapolates the Zn²⁺ results.

It is clear that this new structure has a catalytic core arrangement that is very similar to the available crystal structure. It is not clear what the effect of the new construct is in obtaining/sustaining this, as no control experiments of rate, CD or NMR are reported for the WT two-strand system for comparison. It is clear that the short-armed construct has a better rate performance than the full-length (still capped) construct, so the authors are already working with a “more active” version of their capped system.

To make the claim “single active structure” there would need to be additional structures resolved in the presence of various metal ions especially Pb²⁺ (as a control), and Mg²⁺ that yield the same structure. In its current form, the paper makes a lot of big claims without the experiments to support them. With the experiments that it does report the paper is more appropriate for a more specialized journal.

Beyond the content, there are various aspects that need improving for publication anywhere. Some of them are listed below in order of their appearance in the manuscript.

1. Overall issues with clarity regarding which statements are facts, supported by the experimental data current or previous, and which statements are assumptions or interpretations (and in places extrapolations).

We made an attempt to make this distinction clearer by rewriting parts of the discussion, as well as we added new sections to the SI providing a more thorough data-based justification of some statements e. g. regarding the similarities of the folds induced by the different ions. Moreover, the addition of all the new experimental data makes our discussion much less reliant on comparisons with literature data, which by its nature required more extrapolations due to differences in experimental conditions, DNAzyme variants etc.

2. Lines 77-78. The statement “initial step” is not supported. Currently there is no evidence regarding the order of the chemical steps, nucleophile activation doesn't necessarily need to be the “initial” step.

The expression “initial state” was removed.

3. Lines 95-96: “validated for unaltered catalytic activity”. How is it unaltered? Where's this evidence? There is no comparison to WT two-strand struct. Clearly the rate is different between their short and full-length structures.

As per Reviewer's request the new version of the paper now includes rate comparisons with a full-length, bimolecular DNAzyme construct (we chose the exact construct used previously for X-ray crystallography). The measured rate constants were of the same order of magnitude for the two compared constructs for all tested ions and at all tested concentrations (in general the crystallographic construct was a bit faster than 8-17_short with Mg^{2+} and Pb^{2+} , while being somewhat slower with Zn^{2+}).

4. Lines 152-153: "we will use the numbering specific to this construct". This made it really hard to follow the arguments in the paper as only new residue numbers were given without any context. Authors should include canonical numbers in parentheses and state the role of the residue when necessary (i.e. G7 (G1) upstream from scissile phosphate)

As per Reviewer's request the standard numbering was added in parenthesis (in bold) whenever mentioning specific residues within the catalytic domain of the DNAzyme in the main manuscript text and figures.

In the SI figures featuring regions of NMR spectra, as well as in the new SI text sections describing the NMR data in detail only the numbering specific for our construct is used. This is because the number of assigned peaks in many spectra which would lead to the figures becoming completely illegible if a double numbering was used.

5. Lines 168-170: It's not clear what is meant by this statement. These are 3 base-pairs, 2 of which are non-canonical that form the L-platform motif commonly observed in the active sites of RNA-cleaving nucleic acid enzymes. It is not clear how other than the crystal structures these would have been observed, and what the emphasis of the statement here is. Consider rephrasing. If anything should be emphasized here it's that these base-pairs form the L-platform motif [Gaines RNA 2020. 26: 111-125].

By saying "only observed in the crystal structures" we wanted to give credit to the previous crystallographic study for being the first to report these interactions for the 8-17 DNAzyme, despite preceding two decades of intensive studies of this system. The "only" was not dropped to avoid confusion. The information that similar base pair arrangements are also found in other catalytic nucleic acids and known as the L-platform motif was added to the Introduction when the crystallographic fold is first discussed.

6. Lines 198-201. In its current form it is not convincing that the Na^+ induces the same fold as Zn^{2+} . The CD data are significantly different. There's no discussion of what kind of similarity in the NMR results would yield in what kind of structural similarity.

The mentioned lines do not yet state that Na^+ induces the same fold as Zn^{2+} , only that the observed CD spectral changes are qualitatively similar for the two ions. We stand by this assessment as throughout the entire spectral range the only difference between the two ions is the slightly higher magnitude of the change induced by Zn^{2+} (when one compares the spectra after the transitions are saturated). Similar differences in the magnitude of spectral changes between the ions were also seen in the original 2009 CD study. Regarding the newly added CD-monitored titrations using Mg^{2+} and Pb^{2+} the magnitude of the changes. Yet, once again the CD data are only really used here to demonstrate that a structural transition occurs for each metal and to estimate the K_d of the interaction, and not to conclude that the folded states are the same – that conclusion is based on the analysis of 2D NMR data.

Regarding the discussion of similarities in the NMR results, we did originally refer the reader to tables in the SI that listed all the long-range NOE cross-peaks that define the tertiary fold and whose pattern repeated themselves for both Zn^{2+} and Na^+ . However, we now realize that such an important point should have been discussed more thoroughly and thus have included two new sections in the SI: "NMR spectral patterns observed for the 8-17 short in the structure determination conditions - long-range NOE contacts and uncommon chemical shift values" and "NMR observations for samples containing only a single metal ion type (Na^+ , Mg^{2+} , Zn^{2+} or Pb^{2+})".

7. Lines 214-217. A good example of the concern in #1. Is this an assumption or a statement based on the data seen? (either way clarify, describe). It does not make chemical sense that a displacement of a Na^+ ion (+1 charge with a large ionic radius), with Zn^{2+} ion (+2 charge and small radius), does not induce structural changes. Is it meant that it doesn't affect the overall structure?

Yes, what we meant here is that the replacement of Na^+ for Zn^{2+} should not affect the global tertiary fold, based on our NMR

results regarding the similarities between the global folds induced by the two ions. To clarify this point, we now explicitly refer to “global structure” when describing this experiment. Several sentences leading to the one pointed out by the Reviewer were also rewritten to better express the idea behind the Na⁺ displacement experiment.

8. Lines 305-308. “The fact that for 8-17_{short} enzymatic rate k_{obs} Zn²⁺ does not saturate with the DNAzyme folding, but continues to increase with concentration well Zn²⁺ into the millimolar range (Figure S7), strongly suggests that a different ion – binding with a much lower affinity – is actually the one required for catalysis”. Isn’t this a different behavior than the WT, and might possibly suggest a different mechanism for the current construct?

No, a very similar behavior was also observed already in the original study that reported 8-17 DNAzyme folding based on FRET results (ref. xx), using a full-length, bimolecular construct. As we noted in the discussion already the authors of that paper suggested that two different ions may be involved. Moreover, our newly added data for the “crystallographic” construct show this behavior.

9. Lines 343-345 “Unfortunately, we were not able to test Pb²⁺ binding to the NaCl stabilized folded state in solution, as in these conditions the lead ion would precipitate as PbCl₂, a very weakly soluble salt (K_{so}= 1.7×10⁻⁵ at 20 °C).” This statement is confusing. What are “these” conditions? Would the system not carry out catalysis if Pb²⁺ were added? It also is never explicitly stated what Na-stabilized system is, is it 400 mM NaCl?

By these conditions we mean Na⁺ concentrations at which Na⁺-induced folding is saturated, which does translate to 400 mM or more NaCl.

This phrase is however no longer in the paper, as we managed to study Pb²⁺ binding in different conditions (with no NaCl added).

10. Lines 353-354. “Pb²⁺ only binds to whatever amount of this state is initially present in the conformational ensemble...” This is inconsistent with all the previous functional data. This would result in very a very low %cleavage with Pb compared to Zn, which is not the case [Woohyun Biochemistry 2021. 60: 1909].

We do not believe that to be the case. In our proposed model the apo state (as well as the state assumed by the DNAzyme when micromolar concentrations of Pb²⁺ are present) is an ensemble of the folded and unfolded states in dynamic equilibrium. After the reaction begins the fraction of the folded state would become gradually depleted and thus other DNAzyme molecules would fold to maintain the conformational equilibrium. This process would continue until most of the DNAzyme molecules (or DNAzyme-substrate complexes in case of bimolecular constructs under single turnover conditions) are cleaved, leading to the observed high cleavage yields even if at any given point the population of re folded state is low.

11. Lines 364-366. Providing a compelling structural explanation here is not sufficient. There’s already a chemical explanation for this behavior. The observation in Ref 19 regarding the effect of Na⁺ with Pb²⁺ is not only a structural one, but also a chemical one. Ref 19 found that both the acid-base pK_a’s and the reaction rates were affected by Na⁺ concentration. It was a test of the computational findings of Ref 20, which showed that Na⁺ ions facilitated the active in-line state by bridging the O2’ and the pro-Rp. Without mentioning the background this it seems like their hypothesis is the only explanation currently available for this observation.

The interpretation of the Na⁺ effect on Pb²⁺ catalysis given by the authors of the original paper is now mentioned in the text. Moreover, we have performed similar experiments (even though just at a few Na⁺ concentrations) on the constructs studied here and observed a similar effect, ascertaining that it is relevant also for our systems.

To strengthen our conclusion that the effect is correlated with DNAzyme folding we also performed an additional series of activity assays on samples containing a fixed Pb²⁺ concentration of 50 uM, as well as different concentrations of Mg²⁺ in the range of 0.0-0.5 mM. All the Mg²⁺ concentrations were too small to produce measurable catalytic cleavage within an hour (which we verified in parallel activity assays), yet large enough to induce at least partial folding of the DNAzyme (according to K_d values we measured using CD spectroscopy). We observed a gradual increase of k_{obs} (up to several fold at 0.5 mM Mg²⁺), which we interpret as resulting from the increasing fraction of the folded state being present for Pb²⁺ to use, as described in the manuscript text.

12. Lines 365 – 366. “the catalytic reaction of 8-17 DNAzyme 365 in the presence of Pb²⁺ (which cannot fold the active state on its own”. This is a major misinterpretation. The fact that further folding wasn’t seen in the concentrations tested does not mean that Pb cannot induce further folding. The following segment from Ref 3 clearly states so. “Since the concentrations of Pb²⁺ used in this investigation (0–100 μM) are much lower than those of Zn²⁺ (0–500 μM) and Mg²⁺ (0–20 mM) due to the Pb²⁺ precipitation problem at higher concentrations, we cannot rule out the possibility that the DNAzyme will fold at much higher Pb²⁺ concentrations (if one can solve the precipitation problem).”

After performing additional NMR and CD experiments in presence of Pb²⁺ (as described in the “**General information for Reviewers ...**” above) we now agree with the Reviewer’s statement and the corresponding section of the discussion was rewritten. However, referring to the quotation from ref. 3 while the authors of that paper indeed did not exclude Pb²⁺-dependent folding they also argued that even if it would occur it would be irrelevant to catalysis.

13. Lines: 377-378. “these conclusions can be extended to catalysis in the presence of Zn²⁺ and other divalent 377 metal ions, notably Mg²⁺”. This is not accurate. Mg²⁺ and Pb²⁺ have different mutational signatures for G6 (G22), so clearly something regarding the metal ion binding is different.

We agree with the Reviewer that not all the conclusions formulated using the Pb²⁺-bound crystal structure can be extended to Mg²⁺ and Zn²⁺ due to a likely different coordination geometry of the catalytically relevant metal ion. We never suggested otherwise in the original version of the manuscript, as line 377 taken in its entirety read: “many of these conclusions can be extended to catalysis in the presence of Zn²⁺ and other divalent” and the entire following paragraph discussed which among the conclusions can likely be transferred (those regarding the alpha and gamma catalytic mechanism) and which cannot (those involving the role of the metal ion).

14. Molecular dynamics approach is faulty in the current way it is stated. Lines 471 – 475 “The first five conformers from the NMR bundle were solvated using octahedral box of SPC/E water with a minimum distance between box walls and solute of 10 Å and additional Na⁺ ions were added to neutralize the system. Each system was prepared twice, with one copy containing a Zn²⁺ ion manually introduced before solvation, yielding a total of 10 independent simulation setups. No additional salt was added.”

a. The NMR structures do not have the nucleophile and are therefore inactive. The MD should have been done on the reactive system. It’s one thing to collect NMR data from an inactive system (assumably for signal clarity, although that is also not mentioned) but what is the reasoning behind doing MD on the inactive system?

b. The only ions in the system were the charge neutralizing (counter) ions, which for this small system would be ~36. This means the Na concentration for MD was 0 (zero). One cannot interpret “binding sites” in a simulation with 0 bulk concentration of ions. Na⁺ parameters are the best available ion parameters for molecular simulations and Na⁺ ions can easily exchange and bind to different nucleotides within simulations especially in the microsecond scale, therefore there is no excuse for the lack of a significant concentration bulk Na⁺ ions in these simulations. At least 0.140 M NaCl should have been included. Especially for a system that requires a very high concentration of Na⁺ ions in solution. The fact that two sites were occupied >90% suggests those ions were stuck there. This could very well be due to lack of free floating ions to exchange with.

c. Where was the Zn manually placed? Did it move? Was the total charge of the Zn systems +2? Or were 2 Na⁺ removed to neutralize the charge? Or 2 Cl⁻ were added? All these info need to be provided.

After consideration, we decided to remove the MD simulation results from the manuscript. The main conclusions presented in the paper are based entirely on experimental approaches in which we specialize and which we feel provide a strong justification for the presented claims. Our main reason to attempt MD simulations was to obtain a compelling visual representation of the “structural” and “catalytic” binding sites. We repeated all the MD runs as the Reviewer requested and even with the suggested improved settings (added NaCl, presence of RNA residue at the cleavage site) we obtained results similar to the ones initially presented. However, as indeed one of the sites (the deeper “catalytic” one in our nomenclature) displayed stable coordination and no exchange during multiple 1 us MD runs we agree that discussing its occupancy is meaningless for equilibrium properties of the system. It is still worth noting however, that the Na⁺ ion bound in the other (“structural”) site exchanged with bulk Na⁺ multiple times during the same MD runs and thus its high population is likely not an artifact of insufficient sampling.

Figures:

- All figure legends lack necessary descriptions.

The figure captions were expanded.

- Font sizes are inconsistent. Some inner panels are not legible. Some descriptive figures have very large and very small fonts, making it hard to follow (i.e. having to zoom in and out of the figure to read different sections).

Where possible we changed font sizes to be legible when the full page is viewed (in the legends of CD titration experiments this was however impossible to achieve without the legend becoming too large)

- Figure 6 legend says Zn, but the figure is Na.

The typo was corrected.

REVIEWER COMMENTS

Reviewer #1 (Remarks to the Author):

The authors have revised their manuscript in response to the previous three reviews, where I was Reviewer 1. The resubmission has a tremendous amount of experimental data, including some very important newly added data using several different metal ions. Also the manuscript has been reorganized in light of the new data.

I find the experimental data and explanations to be sound and reasonable. The conclusion (page 14) is remarkable, and sensible based on the data, that Zn²⁺ activates the 8-17 DNAzyme for RNA cleavage only at concentrations where 8-17 is fully folded, while low concentrations of Pb²⁺ activate 8-17 even when most of the DNAzyme is not yet folded. The authors have provided strong data and arguments in favor of this new model (Figure 9), rather than the established model (Figure 1d).

Overall I am satisfied that the authors have addressed the substantive concerns raised in the previous reviews, and therefore the resubmitted manuscript is now suitable for Nature Communications. My congratulations to the authors on their elegant advance on the structure of the 8-17 DNAzyme with different metal ions, especially now including Zn²⁺ and Mg²⁺. This is an important study for understanding 8-17 DNAzyme structure and function, by changing our thinking about the interplay of folding and catalysis for this commonly studied and applied DNAzyme.

A few minor comments on the presentation:

1. On page 9, line 271, "culminating in a suggestion that the structure formed is a Pb²⁺-stabilized G-quadruplex" - I suggest this text would benefit from stating explicitly the authors' proposal that the high-Pb²⁺ structure lacks all Watson-Crick base pairs, including for the DNA arms. This proposal is stated clearly in the SI, page 7, lines 198-200 ("breaking of all Watson-Crick pairs within the molecule"), but it was not evident to me from the main text alone that this is the proposal.

2. In the Abstract, in the following text, I think the word "at" is missing at the bracketed location: "suggest that [at] DNAzyme concentrations used in NMR all these ions induce a similar tertiary fold", where this omitted word initially made the text difficult to understand. There are a few similar omissions elsewhere, such as in the last sentence of the Conclusions. Presumably any such issues will be fixed during copyediting, at least for the main text.

3. Comments on the SI text (pages 3-9): Page 3, line 27, "to" is missing. Page 3, line 28, "adapted" should be "adopted". Page 4, line 59, fix "in within". Page 4, line 80, fix text in parentheses. Page 5, line 102, should be "correlating". Page 5, line 119, missing "of". Page 5, line 129, should be "adopted". Page 6, line 155, fix "were be observed". Page 6, line 169, fix "the for the fold". Page 7, line 200, should be either "in the two DNAzyme arms" or "in the DNAzyme's two arms". Page 7, line 225, "we do not observe neither ... nor ..." should be "we observe neither ... nor ..." (avoid the double negative).

4. The text on page 13, beginning "Regarding the apo state, our 1D NMR spectra...", is a very long unbroken paragraph (lines 331-380). This text might be easier to read when broken into two (or more) shorter paragraphs, such as breaking in line 369. Compare the short lengths of the succeeding paragraphs on page 14.

5. For consistency with the text, the hyphen can be removed from "apo- form" in the images of Figures 1d and 9. Similarly in SI text, page 8, line 264.

Reviewer #2 (Remarks to the Author):

I would like to thank the authors for investing substantial time and resources to address the open questions raised during the first review cycle. The new data and connected new scientific directions make the results much clearer and relevant to the field. My previous concerns have been adequately addressed. Still, the new data raise a few points that require additional attention:

Major points:

1. The authors use their NMR data to map the Zn²⁺ binding site and conclude that G7-N7 and T27-P are central for Zn²⁺ interaction. To strengthen this finding CD measurements of two modified variants i.e. 7-deaza dG7 and T27-PS were carried out. To my understanding, according to the proposed model, both modifications should lead to a strong reduction of the Zn²⁺ interaction. However, the binding (K_d) is only weakly influenced for 7-deaza dG7 and basically no effects are seen for T27-PS. The authors state that this data confirms involvement of the respective atoms in Zn²⁺ binding (line 475). I do not see the link between the new data and this interpretation. Please either include more detailed discussion or modify the statement. For the latter, an explanation of the (unexpectedly) small effects in the data would still be useful. Alternatively, activity data using these modifications could also clarify this aspect.

2. The new data show that the observed structural effects may be sequence specific and are clearly an important addition to the manuscript. While this data may limit the general applicability of the proposed model, the authors, in my opinion, do a good job in openly discussion and not overinterpreting their findings. However, since they show the central features for one specific 8-17 sequence and the data indicate (or at least cannot exclude) that other sequences may differ, I would suggest to adjust the title of the manuscript by replacing the 'absolute' term: "The 8-17 DNAzyme as only a single active structure.." (which may not be the case for other sequences) to a more general term, such as: "The 8-17 DNAzyme can operate in a single active structure regardless of the metal ion cofactor" (not excluding that other 8-17 variants may form an alternative conformation).

Minor points:

3. It is not fully clear, how the structural restrains for the hydrogen bounds in the NMR structure

calculation where obtained. Please include the respective information in the method section, i.e. whether all H-bonds were experimentally observed or if also Watson-Crick pairs were assumed (e.g. for the arms).

4. If not done already, please deposit all NMR assignments, including different metal ions, in the BMRB.

5. Since the different experiments were recorded in considerably different concentrations of the DNAzyme, please include the information about the DNAzyme concentration in all relevant (sub)figures and or respective figure captions (e.g. such as already done for the CD data).

6. From my understanding of the data and deduced models, Pb²⁺ should still induce detectable folding in stabilized FRET samples at sufficiently high Pb²⁺ concentrations. It may be helpful to briefly discuss (e.g. in the respective SI text) if the previous data never included this high ratios before (and what the maximal used Pb²⁺ concentration were) and suggest to the field that (if not included in previous data) this would be a good way to test the model in subsequent experiments.

7. Line 13 in the abstract states that DNAzyme have not reached the clinic. While I understand what is meant by this statement, it circumvents the pioneering work of e.g. sterna therapeutics and others who have succeeded in bringing a number of DNAzymes into clinical phase I and phase II trials. I would suggest rephrasing the sentence, e.g. to "no DNAzyme-based drug has been approved"

8. Overall, the manuscript is very long and contains a large number of figures. In general, the text and illustrations are written in more of a "book/dissertation chapter" style. While the readability of this style is well appreciated by this reviewer, it is rather uncommon for the anticipated high-impact publication and is unlikely to be overly appreciated by the journal's wider readership. I would recommend shortening the text considerably and sharpening the central results, the discussion and the figures (text reduction in the main text by 30-40% and figure rearrangements to about 6 (regular sized) main figures should be well feasible).

Reviewer #3 (Remarks to the Author):

This is a substantially improved manuscript. The authors should be commended for the extensive new experimental results they have included and the improved clarity of their discussion points.

This reviewer additionally would like to explicitly thank the authors for providing experimental evidence that Pb²⁺, in fact, does induce further folding as well.

Most of my previous concerns were satisfactorily addressed. There remain some minor concerns and suggestions listed below, which do not need further review.

1. Abstract and other places; saying "virtually identical" structure reads as a subjective statement as the criteria is not clear, and would benefit including a quantitative value. Consider including rmsd of the active site with respect to the crystal structure, especially in the abstract and introduction.

2. Line 108 “we suggest that the previous...”. Consider replacing “suggest” with “propose”.
3. Figure 3 caption, describe the meaning of the red stars.
4. Line 163, include “shown as red stars”, if that’s what the red stars are placed for.
5. Lines 190-192. State why these base pairs are important (important for catalysis / form the active site etc.). Consider mentioning L-platform here.
6. Line 247. “ion known not to induce 8-17 DNAzyme folding”. Consider stating as “not shown to induce”, as it wasn’t previously tested at concentrations as high as the other metal ions were tested at. Ref 11 had gone up to 0.4 mM, and ref 9 up to only 0.1 mM.
7. Line 298. “DNAzyme variants used in¹¹”. Consider saying “variants used in reference 11” or mentioning the work by authors and using superscripted citation.
8. Table 1. “Dozens” do not seem to be appropriate here, as 94 and 90 in the other rows are technically dozens as well.
9. Authors could consider adding a sentence regarding how structurally bigger difference at the arms might correspond to 10ths of Angstrom changes at the hinge where the active site is.
10. Specifically figures 3, 6, 8: the inlet Kd curves are not legible and should be replotted with bigger fonts to be legible at the size they are being used as the inlet. But in general, if plots are not legible at the size they are being presented, they should be replotted with bigger fonts. This is a recurring issue with some of the plots in the SI as well (specifically Figure S14).

REVIEWER COMMENTS

Reviewer #1 (Remarks to the Author):

The authors have revised their manuscript in response to the previous three reviews, where I was Reviewer 1. The resubmission has a tremendous amount of experimental data, including some very important newly added data using several different metal ions. Also the manuscript has been reorganized in light of the new data.

I find the experimental data and explanations to be sound and reasonable. The conclusion (page 14) is remarkable, and sensible based on the data, that Zn²⁺ activates the 8-17 DNAzyme for RNA cleavage only at concentrations where 8-17 is fully folded, while low concentrations of Pb²⁺ activate 8-17 even when most of the DNAzyme is not yet folded. The authors have provided strong data and arguments in favor of this new model (Figure 9), rather than the established model (Figure 1d).

Overall I am satisfied that the authors have addressed the substantive concerns raised in the previous reviews, and therefore the resubmitted manuscript is now suitable for Nature Communications. My congratulations to the authors on their elegant advance on the structure of the 8-17 DNAzyme with different metal ions, especially now including Zn²⁺ and Mg²⁺. This is an important study for understanding 8-17 DNAzyme structure and function, by changing our thinking about the interplay of folding and catalysis for this commonly studied and applied DNAzyme.

A few minor comments on the presentation:

1. On page 9, line 271, "culminating in a suggestion that the structure formed is a Pb²⁺-stabilized G-quadruplex" - I suggest this text would benefit from stating explicitly the authors' proposal that the high-Pb²⁺ structure lacks all Watson-Crick base pairs, including for the DNA arms. This proposal is stated clearly in the SI, page 7, lines 198-200 ("breaking of all Watson-Crick pairs within the molecule"), but it was not evident to me from the main text alone that this is the proposal.

We have added the information about the breaking the Watson-Crick pairs in DNAzyme arms to the main text, in the discussion preceding the fragment cited by the Reviewer. The fragment now reads:

"This second transition is accompanied by the disappearance of imino proton resonances of Watson-Crick base-pairs (Figure 7c), including those from the DNAzyme's arms, and thus we would like to argue that it is a reflection of the known tendency of Pb²⁺ ions to destabilize helical folds at higher molar ratios by direct coordination of nucleobases, especially guanines.³¹ The rather early onset of this transition for 8-17_{short} is likely related to its very short and G-rich helices. This argument is presented in detail in the SI text (strengthened by additional experimental data presented in Figures 9 and 10), culminating in a suggestion that the structure formed is a Pb²⁺-stabilized G-quadruplex."

2. In the Abstract, in the following text, I think the word "at" is missing at the bracketed location: "suggest that [at] DNAzyme concentrations used in NMR all these ions induce a similar tertiary fold", where this omitted word initially made the text difficult to understand. There are a few similar omissions elsewhere, such as in the last sentence of the Conclusions. Presumably any such issues will be fixed during copyediting, at least for the main text."

The missing words pointed out by the Reviewer were added. The entire text was also attentively read in search of similar errors.

3. Comments on the SI text (pages 3-9): Page 3, line 27, "to" is missing. Page 3, line 28, "adapted" should be "adopted". Page 4, line 59, fix "in within". Page 4, line 80, fix text in parentheses. Page 5, line 102, should be "correlating". Page 5, line 119, missing "of". Page 5, line 129, should be "adopted". Page 6, line 155, fix "were be observed". Page 6, line 169, fix "the for the fold". Page 7, line 200, should be either "in the two DNAzyme arms" or "in the DNAzyme's two arms". Page 7, line 225, "we do not observe neither ... nor ..." should be "we observe neither ... nor ..." (avoid the double negative).

All the errors pointed out by the Reviewer were corrected.

4. The text on page 13, beginning "Regarding the apo state, our 1D NMR spectra...", is a very long unbroken paragraph (lines 331-380). This text might be easier to read when broken into two (or more) shorter paragraphs, such as breaking in line 369. Compare the short lengths of the succeeding paragraphs on page 14.

The long paragraph was split into four shorter ones.

5. For consistency with the text, the hyphen can be removed from "apo- form" in the images of Figures 1d and 9. Similarly in SI text, page 8, line 264.

The hyphen was removed from the figures and the SI text was corrected.

Reviewer #2 (Remarks to the Author):

I would like to thank the authors for investing substantial time and resources to address the open questions raised during the first review cycle. The new data and connected new scientific directions make the results much clearer and relevant to the field. My previous concerns have been adequately addressed. Still, the new data raise a few points that require additional attention:

Major points:

1. The authors use their NMR data to map the Zn²⁺ binding site and conclude that G7-N7 and T27-P are central for Zn²⁺ interaction. To strengthen this finding CD measurements of two modified variants i.e. 7-deaza dG7 and T27-PS were carried out. To my understanding, according to the proposed model, both modifications should lead to a strong reduction of the Zn²⁺ interaction. However, the binding (K_d) is only weakly influenced for 7-deaza dG7 and basically no effects are seen for T27-PS. The authors state that this data confirms involvement of the respective atoms in Zn²⁺ binding (line 475). I do not see the link between the new data and this interpretation. Please either include more detailed discussion or modify the statement. For the latter, an explanation of the (unexpectedly) small effects in the data would still be useful. Alternatively, activity data using these modifications could also clarify this aspect.

The observed effect of the 7-deaza dG7 is a fivefold decrease of the binding affinity, which in our opinion is substantial enough to confirm the involvement of the dG7-N7 atom in the interaction. We do not necessarily state that this atom is a first sphere ligand of the metal ion - as in this case a stronger

effect might indeed be expected - but it can for example be involved in second sphere contacts with Zn²⁺ bound water molecules.

In case of the T27-PS modification an additional complication arises due to the sample being a mixture of two diastereoisomers. If only one phosphate oxygen atom is involved in the interaction (which is common in metal binding sites in nucleic acids), the measured effect of the modification would be attenuated by the presence of the second diastereoisomer with unaffected K_d. As described in the SI section "Analysis of CD-titrations of phosphorothioate (PS) modified constructs" depending on whether we consider this complication or not when fitting the CD data we obtain either a two- or fourfold decrease in binding affinity. Such a change is quite small, but likely significant as shown by comparison to the T7-PS modification for which no apparent change in K_d can be revealed regardless of the method of fitting the data. The effect of T27-PS modification is indeed only very weakly pronounced upon visual inspection of the CD data as compared that of 7-deaza dG7, but the measured changes do affect the K_d fitting.

We realize that all these important considerations were glossed over in the original manuscript and thus we expanded the appropriate section of the text to read:

"The involvement of G7(N1)-N7 and T27(N12)-P in Zn²⁺ binding was tested by studying the folding of 8-17_short variants containing either the 7-deaza or phosphorothioate modifications at the respective sites (see Figure S14 and SI text). For the 7-deaza-G7(N1) construct a fivefold decrease in binding affinity was observed, while for the T27(N12) phosphorothioate modification the decrease was two- to fourfold, depending on the method of data fitting (see SI text). These K_d changes confirm the involvement of these two moieties in the interactions, yet their magnitudes may point towards second sphere interactions, rather than direct metal coordination."

2. The new data show that the observed structural effects may be sequence specific and are clearly an important addition to the manuscript. While this data may limit the general applicability of the proposed model, the authors, in my opinion, do a good job in openly discussion and not overinterpreting their findings. However, since they show the central features for one specific 8-17 sequence and the data indicate (or at least cannot exclude) that other sequences may differ, I would suggest to adjust the title of the manuscript by replacing the 'absolute' term: "The 8-17 DNAzyme as only a single active structure.." (which may not be the case for other sequences) to a more general term, such as: "The 8-17 DNAzyme can operate in a single active structure regardless of the metal ion cofactor" (not excluding that other 8-17 variants may form an alternative conformation).

The title was changed according to the Reviewer's suggestion.

Minor points:

3. It is not fully clear, how the structural restraints for the hydrogen bounds in the NMR structure calculation were obtained. Please include the respective information in the method section, i.e. whether all H-bonds were experimentally observed or if also Watson-Crick pairs were assumed (e.g. for the arms).

The following phrase was added to the Methods section: "Hydrogen bond restraints for G-C base pairs were imposed based on the observation of strong NOE cross-peaks between specific cytidine NH₂ and guanosine imino protons. For the C22-G28 base pair the assignment of the imino proton was confirmed using site-specific isotopic labelling (Figure S5a)."

4. If not done already, please deposit all NMR assignments, including different metal ions, in the BMRB.

The NMR assignments for additional metal ions were deposited in BMRB under the number 52355.

5. Since the different experiments were recorded in considerably different concentrations of the DNAzyme, please include the information about the DNAzyme concentration in all relevant (sub)figures and or respective figure captions (e.g. such as already done for the CD data).

The concentration information was included directly in NMR-related Figures 3, 6 and S4. For kinetic assays-related Figures 10, S2 and S3 this information was added to the figure captions.

6. From my understanding of the data and deduced models, Pb²⁺ should still induce detectable folding in stabilized FRET samples at sufficiently high Pb²⁺ concentrations. It may be helpful to briefly discuss (e.g. in the respective SI text) if the previous data never included this high ratios before (and what the maximal used Pb²⁺ concentration were) and suggest to the field that (if not included in previous data) this would be a good way to test the model in subsequent experiments.

This is indeed a very interesting question. While the previous studies never approached as high absolute Pb²⁺ concentrations as we did here, the Pb²⁺-to-DNA concentration ratios used is some of them were actually much higher than what we reached here. For example, in the bulk FRET study the DNAzyme concentrations were <100 nM while the Pb²⁺ concentrations were up to 100 μM (over 1000-fold molar excess). We thus find it not impossible that the molar excess was actually too high to observe Pb²⁺-induced folding to the active state due to the ability of Pb²⁺ to disrupt helical conformations at such high molar excess. We added a brief paragraph to the SI text introducing this idea:

"One additional factor to consider when discussing the lack of Pb²⁺-induced folding in the FRET experiments is that these studies employed very high Pb²⁺-to-DNA molar ratios. In the bulk FRET study the DNAzyme concentration was <100 nM with Pb²⁺ concentration reaching up to 100 μM, yielding over thousandfold molar excess. In the smFRET the exact DNAzyme concentration was not reported, yet it was certainly submicromolar (the smFRET samples were diluted from 1 μM stocks) with 20 μM Pb²⁺, yielding at least a twentyfold excess. At such high molar ratios Pb²⁺ might actually interfere with proper DNAzyme folding, as discussed above in the "*The structure forming at higher Pb²⁺-to-DNA molar ratios during 8-17_short titrations with Pb²⁺*" section."

7. Line 13 in the abstract states that DNAzyme have not reached the clinic. While I understand what is meant by this statement, it circumvents the pioneering work of e.g. sterna therapeutics and others who have succeeded in bringing a number of DNAzymes into clinical phase I and phase II trials. I would suggest rephrasing the sentence, e.g. to "no DNAzyme-based drug has been approved"

The abstract was changed according to the Reviewer's suggestion.

8. Overall, the manuscript is very long and contains a large number of figures. In general, the text and illustrations are written in more of a "book/dissertation chapter" style. While the readability of this style is well appreciated by this reviewer, it is rather uncommon for the anticipated high-impact publication and is unlikely to be overly appreciated by the journal's wider readership. I would

recommend shortening the text considerably and sharpening the central results, the discussion and the figures (text reduction in the main text by 30-40% and figure rearrangements to about 6 (regular sized) main figures should be well feasible).

We appreciate the Reviewer's suggestion and understand the benefits a more streamlined version of the manuscript would provide. However, after attempting a similar rewrite we realized that it would require us to relegate many important details to the SI where they would be lost among already very abundant SI text sections. As the paper reports a large number of new and unexpected results we think that the current somewhat longer format might be best fitted to properly present them.

Reviewer #3 (Remarks to the Author):

This is a substantially improved manuscript. The authors should be commended for the extensive new experimental results they have included and the improved clarity of their discussion points.

This reviewer additionally would like to explicitly thank the authors for providing experimental evidence that Pb²⁺, in fact, does induce further folding as well.

Most of my previous concerns were satisfactorily addressed. There remain some minor concerns and suggestions listed below, which do not need further review.

1. Abstract and other places; saying "virtually identical" structure reads as a subjective statement as the criteria is not clear, and would benefit including a quantitative value. Consider including rmsd of the active site with respect to the crystal structure, especially in the abstract and introduction.

The expression "virtually identical" was changed to a weaker one: "very similar" in the abstract and Introduction and the RMSD value for the catalytic domain was included in the abstract.

2. Line 108 "we suggest that the previous...". Consider replacing "suggest" with "propose".

The expression was changed according to the Reviewer's suggestion.

3. Figure 3 caption, describe the meaning of the red stars.

The meaning of the red stars (resonances belonging to a different spectra form) was included in Figure 3 caption.

4. Line 163, include "shown as red stars", if that's what the red stars are placed for.

The proposed inclusion was made.

5. Lines 190-192. State why these base pairs are important (important for catalysis / form the active site etc.). Consider mentioning L-platform here.

The following phrase was added: "These base pairs are crucial for the formation of the "L-platform"²³ tertiary fold in the catalytic domain and for positioning G29(G14) base to act as a general base during catalysis."

6. Line 247. "ion known not to induce 8-17 DNAzyme folding". Consider stating as "not shown to induce", as it wasn't previously tested at concentrations as high as the other metal ions were tested

at. Ref 11 had gone up to 0.4 mM, and ref 9 up to only 0.1 mM.

The proposed change was made.

7. Line 298. "DNAzyme variants used in11". Consider saying "variants used in reference 11" or mentioning the work by authors and using superscripted citation.

The expression was changed to "variants used in reference 11".

8. Table 1. "Dozens" do not seem to be appropriate here, as 94 and 90 in the other rows are technically dozens as well.

The expression in the table was changed from "dozens μM " to "likely $<100 \mu\text{M}$ " and the basis for such an estimate was added to the table footnote.

9. Authors could consider adding a sentence regarding how structurally bigger difference at the arms might correspond to 10ths of Angstrom changes at the hinge where the active site is.

We thank the Reviewer for this suggestion, however after consideration we were not able to find a good place to organically include a similar phrase in the current flow of the manuscript.

10. Specifically figures 3, 6, 8: the inlet Kd curves are not legible and should be replotted with bigger fonts to be legible at the size they are being used as the inlet. But in general, if plots are not legible at the size they are being presented, they should be replotted with bigger fonts. This is a recurring issue with some of the plots in the SI as well (specifically Figure S14).

The inlets in Figures 3, 6 and 8 were redone with larger fonts. The different panels of Figure S14 were also replotted with similar changes. Figure S2 was also corrected to properly show an axis legend that was previously partially obstructed by another panel.

REVIEWERS' COMMENTS

Reviewer #2 (Remarks to the Author):

Thank you for clarifying my remaining points and congratulations to the nice work!